# Incremental Element Deletion-Based Finite Element Analysis of the Effects of Impact Speeds, Fall Postures, and Cortical Thicknesses on Femur Fracture

**DOI:** 10.3390/ma15082878

**Published:** 2022-04-14

**Authors:** Yangyang Cui, Dingding Xiang, Liming Shu, Zhili Duan, Zhenhua Liao, Song Wang, Weiqiang Liu

**Affiliations:** 1Tsinghua Shenzhen International Graduate School, Tsinghua University, Shenzhen 518055, China; cuiyy20@mails.tsinghua.edu.cn (Y.C.); dzl20@mails.tsinghua.edu.cn (Z.D.); 2School of Mechanical Engineering and Automation, Northeastern University, Shenyang 110057, China; 3State Key Laboratory of Tribology, Department of Mechanical Engineering, Tsinghua University, Beijing 100084, China; liaozh@tsinghua-sz.org; 4Key Laboratory of Biomedical Materials and Implant Devices, Research Institute of Tsinghua University in Shenzhen, Shenzhen 518057, China; 5Department of Mechanical Engineering, School of Engineering, The University of Tokyo, Tokyo 1138656, Japan; l.shu@mfg.t.u-tokyo.ac.jp

**Keywords:** fracture, femur, finite element analysis, numerical simulation, falling parameters

## Abstract

The proximal femur’s numerical simulation could give an effective method for predicting the risk of femoral fracture. However, the majority of existing numerical simulations is static, which does not correctly capture the dynamic properties of bone fractures. On the basis of femoral fracture analysis, a dynamic simulation using incremental element deletion (IED)-based finite element analysis (FEA) was developed and compared to XFEM in this study. Mechanical tests were also used to assess it. Different impact speeds, fall postures, and cortical thicknesses were also studied for their implications on fracture types and mechanical responses. The time it took for the crack to shatter was shorter when the speed was higher, and the crack line slid down significantly. The fracture load fell by 27.37% when the angle was altered from 15° to 135°, indicating that falling forward was less likely to cause proximal femoral fracture than falling backward. Furthermore, the model with scant cortical bone was susceptible to fracture. This study established a theoretical foundation and mechanism for forecasting the risk of proximal femoral fracture in the elderly.

## 1. Introduction

Proximal femoral fracture is an important cause of disability and death around the world, especially in the elderly [1]. The number of patients with proximal femoral fractures continues to grow every year; there is currently one proximal femoral fracture every 3 s in the world [2]. The disability rate of proximal femoral fracture is more than 50%, and 8–10% of patients die within 30 days because of proximal femoral fracture [3]. Even surviving patients are often accompanied by various complications that seriously affect their quality of life. Only 40–60% of them are likely to recover their pre-fracture level of mobility [4]. Pekka et al. [5] pointed out that about 90% of proximal femoral fractures are caused by falls, and the directions and speeds of the falls are the main factors that affect the proximal femoral fractures. However, not all falls can cause proximal femoral fractures. Compared with the elderly, falls, which occur in young and middle-aged people in the same way and with the same external factors, rarely cause similar fractures. This may be related to the thickness of the cortical bone with age [6]. Heikki et al. [7] also pointed out that proximal femoral fractures are related to different fall conditions (fall postures, impact speeds) and the thicknesses of cortical bones from a biomechanical perspective. Therefore, a comprehensive understanding of the proximal femoral fracture factors such as fall postures, impact speeds, and the thicknesses of femoral cortical bones needs to be studied.

Bone fracture may now be studied at both the micro and macro scales due to advances in computer modeling. With a correlation of nearly 90%, linear finite element models have been effectively used for the prediction of the elastic response and fracture load of a human femur [8]. Despite efforts to simulate human femur behavior, fracture patterns have only been modelled using numerical methodologies on a few occasions. Some studies have focused on proximal fracture simulation, with the majority of them yielding short fracture pathways [9,10,11,12,13]. Li et al. [9] conducted a three-point bending mechanical test on a cortical bone sample with a single-edge incision. Based on the result of the test, an extended finite element method (XFEM) was developed and used to analyze its deformation and cracks. Alba et al. [10] also put forward XFEM for crack growth without remeshing. Giambini et al. [11] studied the feasibility of predicting the risk of vertebral fracture by XFEM. The results showed that the crack propagation paths based on XFEM are in good agreement with the experimental observations. However, XFEM can only simulate the initial position of a crack and extremely small cracks due to convergence issues [12,13]. The extant numerical simulations are mostly static ones, which have difficulty representing the dynamic crack expansion process [8,14].

Therefore, in this study, the dynamic simulation by incremental element deletion (IED)-based finite element analysis (FEA) on femoral fracture analysis was developed and compared with XFEM, which was also evaluated with mechanical tests. This developed method was further used to simulate the effects of different impact speeds, fall postures, and cortical thicknesses on fracture types and mechanical responses. It potentially provides a trustworthy method for indicating the proximal femoral fracture risk in the elderly, which can serve as a theoretical foundation for clinical orthopedics and surgical procedure evaluation.

## 2. Materials and Methods

Five newly designed fourth-generation composite femurs provided by Sawbones (Sawbones, Pacific Research Laboratories, Vashon, WA, USA) were used to investigate fracture behaviors. In the literature [15,16], the composite femur has been widely employed as a substitute for actual bone. It is vital to note that these specimens are intended to mimic the biomechanical qualities of young, healthy femurs. Axial compression, bending, and torsion tests were used to assess these commonalities, with the related stiffness and ultimate failure strength measured [15,16]. Artificial bone has advantages for model validation since it eliminates the variety of properties found in biological tissues. Because of their uniform qualities in two separate zones, smooth surface, and minimal variability between specimens, composite bones are excellent for developing controlled analysis [15]. These composite models’ failure patterns are similar to those reported for human bones [16]. The composite femurs were scanned by computerized tomography (CT) at first. The entire composite femur was scanned by CT with a slice thickness of 2.5 mm and a pixel width of 0.938 mm (GE MEDICAL SYSTEMS/LightSpeed 16 Computed Tomography Scanner System, 80 kV, 443.52 mAs, 512 × 512 matrix, 52 images). IED-based FEA on femoral fracture analysis was developed and compared with XFEM, which was also evaluated by mechanical tests. The strains of 8 positions on the surfaces of composite femurs were collected in elastic tests (as shown in Figure 1a–d). The fracture loads and crack propagation paths of composite femurs were collected in fracture tests. 

### 2.1. Mechanical Tests

Mechanical tests were performed on the composite proximal femurs. The strain was measured in the elastic regime at 8 positions on the surfaces of composite femurs (as shown in Figure 1d, e. The composite femur was fixed by a disposable fixation container in frame 3, and then it was put into fixture 2, and finally fixed on the mechanical testing machine by the nut of frame 1. Next, the load was increased until bone failure in the fracture test. Then the crack propagation path and fracture load were obtained. Two angles (α and β) were defined to describe the placement direction of the composite femur in the test and fall posture in the numerical simulation, where the angle α was 0–135° with reference to the long axis of the femur in the frontal plane, and β was 0–45° with reference to the femoral neck axis in the horizontal plane, as shown in Figure 1a,b. The distal end of the composite proximal femur was fixed with a clamp. The α angle of 10° and β angle of 15° were used to represent the placement direction of human, as reported by other authors [17,18]. Furthermore, point A (Figure 1a) was the central position of the spherical region surface with a 35 mm diameter on the femoral head.

The composite femur was tested on a 10 kN universal hydraulic testing machine (Instron 8801, load cell 10 kN). As shown in Figure 1c, the distal end of the composite femur was inserted into the dental powder mixture to maintain the composite femur with an α angle of 10° and β angle of 15°, and it was fixed for about 30–60 min. The loads were applied to the femur in the direction of stance loading (Figure 2). Because it matches the physiological plan of a standing-up human position, this is the most commonly studied configuration in the literature [19]. During the test, 8 strain gauges (Qinhuangdao Aifu Te Electronic Technology Co., Ltd., Hebei, China, BX-120) were used to measure the strains at 8 different positions on the surfaces of the composite femurs. The specific locations where the strain gauges were pasted are shown in Figure 1e. To maintain quasi-static conditions, the load was increased to the different values (250 N, 500 N, and 750 N with an actuator speed of 0.3 mm/s). This test confirmed the linear elastic behavior of the femur, as described by another author [20].

### 2.2. Numerical Simulation

#### 2.2.1. Mesh Convergence Analysis

Mimics (Materialise Inc., Leuven, Belgium) was used to construct a finite element (FE) model of the composite proximal femur. Abaqus (Simulia Inc., Providence, RI, USA) was used to divide the model of the composite proximal femur into two three-dimensional (3D) solid parts. Because of the different materials of the composite femurs, the model was endowed with two material properties, i.e., cancellous bone and cortical bone. The loading conditions of the simulation were exactly the same as the mechanical tests. Similarly, the strains at 8 points on the surfaces in the FE model were calculated when the loading force was 250 N, 500 N, and 750 N. In addition, the fracture load and the crack propagation path were calculated by IED.

The element size of the FE model is very important for the accuracy of the result [21]. Therefore, it is necessary to investigate the influences of different tetrahedral element sizes on the FE results to select an appropriate element size. In this study, both 3D solid parts were divided into 3D tetrahedral mesh, and the element sizes were set to 0.5 mm, 0.75 mm, 1 mm, 1.5 mm, 2 mm, 2.5 mm, and 3 mm. The composite proximal femur models with 3D tetrahedral mesh and materials are shown in Figure 2a,b. The cortical bone was transparent, and the internal opaque part was cancellous bone. The parameters required in the both numerical methods including Young’s modulus, Poisson’s ratio, bone density, and compressive failure strain were obtained according to the study by Marco et al. [22].

The boundary conditions and loading state were set to be as close to the experimental conditions as possible. The distal part of the diaphysis was restrained to the testing rig, which simulates the surgical cement embedment, and the force was imparted in the right direction, as in the tests. The distal end of the composite femur was fixed. An α angle of 10° and β angle of 15° were used to simulate the fall posture (as shown in Figure 2c). The loading was exerted on a spherical region with a 35 mm diameter on the femoral head (this is in agreement with the numerical models developed by [20]), and the fracture process was simulated. The maximum principal stress, displacement at point A in Figure 1a, and the running time of the program were recorded in the models with 7 different element sizes mentioned above. Finally, the results of FEA were compared with the results of mechanical tests.

#### 2.2.2. IED-Based FEA

IED-based FEA on the femoral fracture analysis were developed. It was carried out through a Python script that interacted with Abaqus; each crack propagation was regarded as a new iteration process. The crack propagation path and fracture load with time during the impact process were obtained. In this study, Young’s modulus, bone density, and Poisson’s ratio were assigned to the model according to the study by Marco et al. [22]. An initial increment “n” of the load was set, and the principal strain of the composite femur was calculated. Furthermore, the ratio of the tensile failure strain to the compressive failure strain of the composite femur material was set to 0.6 according to the literature [23]. According to the compressive failure strain in the study by Marco et al. [22] and the calculated tensile failure strain, it was judged whether the element was invalid. Then the dynamic load continued to increase with time. During the loading process, the principal strain of each element in the composite femur model was compared, and when it exceeded the failure strain, Young’s modulus was reduced to the minimum value (E = 1 MPa) in order to reduce the element stiffness to a negligible value. This technique improved the deformation problem when deleting elements. Therefore, the elements remained in the model with negligible stiffness. The scheme of the automatized process is also illustrated in Figure 3.

XFEM was also used for comparison with IED-based FEA in the femoral fracture analysis. By using the XFEM module in Abaqus/Standard, the virtual crack closure technology (VCCT) was used to simulate the crack propagation process. The load was continuously increased until the composite femur failed to fracture. The crack propagation path and fracture load of the composite femur were predicted. The Young’s modulus, bone density, Poisson’s ratio, and compressive failure strain required in this method were conducted according to the study by Marco et al. [22]. The critical energy value (G_C_), necessary for XFEM to predict the start of crack propagation, was estimated from the fracture toughness K_C_, as shown in Equation (1), and the fracture toughness K_C_, was related to bone density and could be obtained by Equation (2). The following expressions determine these relationships [24]:(1)$$G_{C}\left({\mathrm{Jm}}^{-2}\right)=\frac{K_{c}^{2}\left(1-\nu^{2}\right)}{E}$$
(2)$$K_{c}\left({\mathrm{Nm}}^{-1.5}\right)=0.7413\times 10^{6}\times \rho^{1.49}$$

#### 2.2.3. The Related Factors of Fracture

To understand falling risk in the elderly, the developed method was used to simulate the effects of different impact speeds, fall postures, and cortical thicknesses on fracture types and mechanical responses. Because gravitational acceleration (1 g = 9.81 m/s^2^) exists in daily life, it was included in the model to simulate the actual falls, with an initial velocity of 3170 mm/s (average femur impact velocity [25]). Thus, in this study, 4 impact speeds (1000 mm/s, 2000 mm/s, 7000 mm/s, 14,000 mm/s) were simulated, and the distal end of the composite femur was completely fixed. The α angle of 90° and β angle of 30° were used to simulate the fall posture, and the dynamic pressure loading was applied perpendicular to the direction of the composite femur head.

A total of 108 fall postures were described by the two angles defined above (as shown in Figure 2a,b). The loading conditions set for the 108 models were the same as the 4 speed simulations. Dynamic pressure loading was applied at a speed of 0.3 mm/s on a 35 mm diameter spherical region above the femoral head. It was performed until the composite femur fractured. Annur et al. [26] pointed out that cortical thickness index (CTI) was a significant risk factor for proximal femoral fracture. Thus, in order to understand the effects of cortical bone thickness on femoral fracture in clinics, composite proximal femur models with 8 different cortical thicknesses were simulated. Material properties of cancellous bone and cortical bone were assigned to the different number of elements. The percentages of cortical bone mass/bone mass were equivalent to the CTI, which were set to 99%, 80%, 60%, 40%, 20%, 10%, and 1% to represent different cortical thicknesses. A hollow model was set up for comparison. The distal end of the composite femur was completely fixed. An α angle of 90°and β angle of 30° were used to simulate the fall posture. The dynamic pressure loading was applied at a speed of 0.3 mm/s on a 35 mm diameter spherical region above the femoral head. It was performed until the composite femur was fractured.

## 3. Results

### 3.1. Mesh Convergence Results

Figure 4a shows the effect of element sizes on the ratio of simulated displacement to experimental displacement of position A. It could be seen that the larger elements gave poor results, and smaller elements had more accurate results. Thus, the element size should be less than or equal to 1.0 mm. It can be seen from Figure 4b that in order to reduce the influence of element size on the maximum principal stress of the composite femur and improve the accuracy of FEA, the element size should be less than 2 mm. According to the analysis above, the element size should be selected within the range of 1.0 mm or less. Good predicted accuracy was found on a fine mesh, while a much longer calculated time was determined (as shown in Figure 4c). It was found that the running time was about 1.5 times longer than that of the 1 mm element size when the element size was within the range of 1.0 mm. Therefore, 1 mm was selected as the appropriate element size in this study.

### 3.2. Comparison of Both Numerical Simulations

#### 3.2.1. Fracture Load and Crack Propagation Path

The relative errors between the simulated and experimental fracture load calculated by Equation (3) are listed in Table 1. The crack propagation path of the fracture test is shown in Figure 5c, while the corresponding numerical results are shown Figure 5a,b for each method evaluated. It can be seen in Figure 5a,b that the projected fracture location and early phases of crack propagation were identical to the failure zone seen in Figure 5c. Furthermore, the numerical results were also confirmed by other researchers using composite femurs [27]. The experimental test revealed a maximum load of 6573 ± 10 N in terms of fracture loading. The fracture test was performed on the neck, as described by other authors [28]. Our numerical simulations similarly anticipated failure in this zone; see Figure 5d for the projected fracture route for the stance loading case. It was found that the results including relative error and crack propagation path of IED-based FEA were closest to the results of mechanical tests. Therefore, in the following study, IED-based FEA was selected for simulation of different impact speeds, fall postures, and cortical thicknesses on fracture types and mechanical responses.
(3)$$\delta =\frac{\left|\mathrm{Fmax}1-\mathrm{Fmax}2\right|}{\mathrm{Fmax}1}\times 100\%$$

δ: Relative error,Fmax1: Experimental fracture loadFmax2: Simulated fracture load

**Table 1 materials-15-02878-t001:** The relative error between the simulated and the experimental fracture load.

Mechanical Responses	Fracture Test	Numerical Simulation	
		XFEM	IED
Fracture load (N)	6573 ± 10 N	5897	6345
Relative error	0%	10.2%	3.4%

**Figure 5 materials-15-02878-f005:**
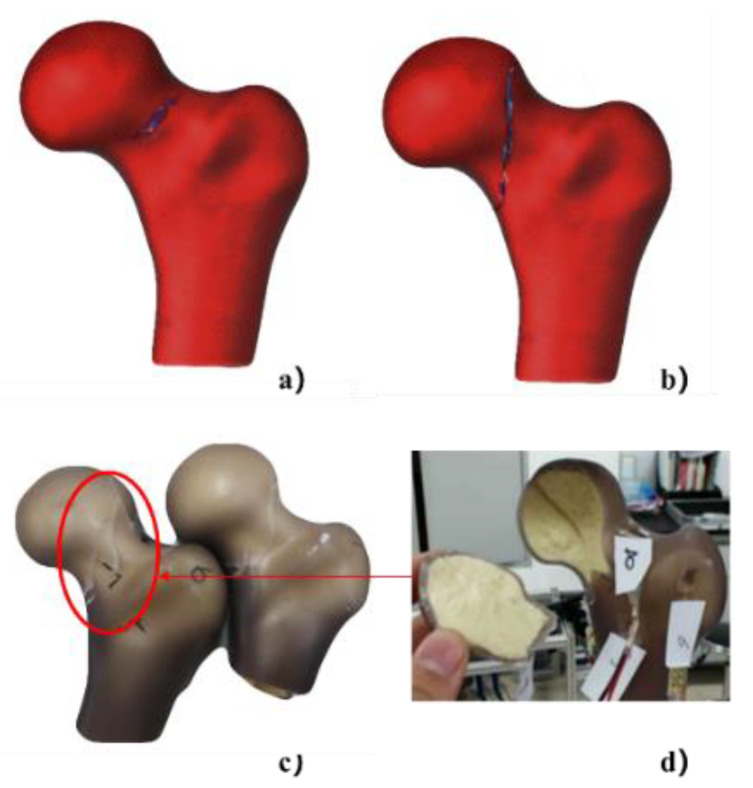
Crack propagation paths obtained by simulation and mechanical tests: (**a**) XFEM method; (**b**) IED-based FEA method; (**c**) mechanical tests; (**d**) broken composite femur after fracture test.

#### 3.2.2. Analysis in Elastic Regime

From the above results in Section 3.2.1, we calculated the fit curve of the IED-based FEA. The generated models accurately represented the mechanical behavior of composite femur loading, as seen in the experiments. Femurs were fitted with eight strain gages in the elastic regime. Because the three measurements were taken at three different loading levels (250 N, 500 N, and 750 N), each femur had a total of 24 strain data points to validate the numerical model. It can be seen that numerical model predictions were correlated (R^2^ = 0.9173) to the experimental values for 24 strains (8 strain gages for each load case) at different loads (250 N, 500 N, 750 N) (as shown in Figure 6). The values for the mean and standard deviation in the experimental and numerical results were, respectively, 52.85 and 246.27, and 64.37 and 237.52. There was a significant correlation between the experimental and numerical strain (COV = 58,196 > 0, 95%CI, 0.79–1.05, *p* < 0.0001).

### 3.3. Simulation Results of Fracture-Related Factors

#### 3.3.1. Effects of Impact Speeds

Figure 7 shows the effects of impact speeds on the mechanical responses of composite proximal femoral fractures. Figure 7a shows the effects of different speeds on fracture loads. It could be seen that the fracture load decreased with the increase of the impact speed, which meant that the fracture loads of the composite proximal femurs were negatively correlated with the impact speeds. Figure 7b shows the effects of speed on fracture time. It could be seen that with the increase of impact speed, the fracture time became shorter. Figure 7c shows the effects of speed on the crack lengths. It could be seen that when the impact speed exceeded a certain range, greater was the impact speed and longer was the crack lengths. It could be seen that when the impact speed exceeded a certain range, the impact speed was greater, the crack length was longer in the composite femur. Figure 7d shows the effects of impact speeds on the instantaneous speeds, which means the speed of crack propagation of composite proximal femoral fractures. It could be seen that with the increase of impact speed, the instantaneous speed at the time of fracture became greater, with greater harm to the human body.

Figure 8 shows the simulated crack propagation paths of the composite proximal femur under different impact speeds. It could be seen that all four impact speeds caused femoral neck fractures, and all cracks started from the femoral neck. As the load increased, the crack gradually extended, and finally the entire composite femoral fracture occurred. It could also be seen that the crack paths caused by different impact speeds were different. As the impact speed increased, the fracture line gradually thickened from the upper side of the composite femur neck and moved downward to the greater trochanter of the composite femur.

#### 3.3.2. Effects of Fall Postures

When β = 0°, 15°, 30°, and 45°, the relationships between the α angle and the maxi-mum principal stress, maximum principal strain, and fracture time were analyzed. The results are shown in Table 2. It was found that when β = 0°, 30°, 45°, the α value had no significant correlation with the maximum principal stress (*p* > 0.05), maximum principal strain (*p* > 0.05), or fracture time (*p* > 0.05). When β = 15°, the α angle was significantly correlated with the maximum principal stress of the model (R^2^ = 0.887, *p* < 0.05). When β = 0°, 30°, 45°, the α angle was significantly correlated with the maximum principal strain of the model (R^2^ = 0.913, *p* < 0.05). However, the α angle had no significant correlation with the model fracture time (R^2^ = 0.689, *p* > 0.05) and the maximum principal stress (R^2^ = 0.776, *p* > 0.05). When β = 15° and α = 15°, the maximum principal stress and the maximum principal strain of the composite femur model were the smallest. In this case, the fracture rarely occurred, and the fracture load was 7321 N. When β = 15° and α = 135°, the maximum principal stress and the maximum principal strain of the model were the largest. In this case, the fracture was most likely to occur, and the fracture load was 5317 N.

When β = 15°, the maximum principal strain of the composite femur decreases with the increase of α angle. The more easily the element failure occurred, and the greater the possibility to fracture. Through iterative calculations, the composite femur model would break as the number of negligible elements reached a certain number. Thus, the smaller the maximum principal strain, the less prone to fracture. In other words, falling forward was less likely to cause a composite proximal femoral fracture than falling backward. Eugenio et al. [29] found that for falling backward, if one could rotate the body forward or sideways, the risk of proximal femoral fracture could be reduced, which was consistent with the numerical simulation results of this study. In addition, due to the presence of vascular perforation holes on the outer and upper sides of the femoral neck, the bone cor-tex was thinner, and the bone mass loss rate was faster than the inner and lower sides [30]. When the fall occurred, the load on the outer and upper sides of the femoral neck was compressed, while the femoral neck was under tension on the inner and lower sides. Thus, a shear force was formed at the femoral neck in instantaneous time, which made the composite proximal femur more likely to cause femoral neck fracture. Figure 9 shows typical fracture crack propagation paths under different fall postures. It could be seen that different fall postures would lead to two different fracture types: composite femur neck fracture and intertrochanteric fracture. The composite femoral fracture position was at the top of the femoral neck (Figure 9a–d); however, the composite femoral fracture position was at the bottom of the femoral neck (Figure 9e,f). In addition, the fracture position was at the upper end of the femoral intertrochanteric (Figure 9g,h). When α angle did not change, as β increased, the fracture axis moved downward (Figure 9d–f) from the top of the composite femur neck to the bottom of the composite femur neck.

#### 3.3.3. Effects of Cortical Thicknesses

The mechanical responses of the models with 8 different cortical thicknesses are listed in Table 3. It could be seen that the maximum principal stress and the maximum principal strain of the composite femur model with the cortical bone mass/bone mass of 99% were 140.6 MPa and 0.994 × 10^−2^, respectively, and the fracture load was 7329 N. The maximum principal stress and the maximum principal strain of the composite femur model with the cortical bone mass/bone mass of 0% were the largest in the 8 models, which were 397.9 MPa and 1.301 × 10^−2^, respectively, and the fracture load was 4575 N. It showed that the thicker the proximal cortical bone of the composite femur, the smaller the maximum principal strain of the model.

Figure 10 shows the crack propagation paths of the cortical bone models with different thicknesses. It could be seen that the cortical thicknesses had great effects on the fracture types and mechanical responses. As the cortical bone became thinner, the fracture line gradually moved from the composite femoral neck to the intertrochanter, resulting in different types of composite proximal femoral fractures, which indicated that the thickness of cortical bone was an important factor affecting the types of composite proximal femoral fractures. It could be seen that when the model contained cortical bone, the crack began to initiate from the femoral neck, and it gradually extended upward. Furthermore, when the entire model was cancellous bone, the femoral head was fractured.

## 4. Discussion

IED-based FEA resulted in longer fracture trajectories than XFEM. Due to convergence issues, XFEM produced unsatisfactory results for long crack trajectories. As a result, pathways generated using an IED-based FEA showed good convergence behavior, resulting in extended trajectories. Because each increment of the fracture growth represents a fresh simulation, this technique avoids convergence issues. When comparing the two procedures, element elimination has more issues due to the presence of distorted elements, which might cause the numerical process to slow down. As a result, the IED-based FEA technique produces the best results in terms of convergence and fracture path length, and it can be applied to a variety of loading scenarios.

The results of this study showed that the fracture types, fracture loads, and crack propagation paths of composite proximal femurs were significantly affected by the fall postures, impact speeds, and cortical thicknesses. As the impact speed increased, the complexity and roughness of fracture cracks also increased, which was consistent with the results in the literature [26,31]. Ren et al. [32] pointed out that when the load continues to increase beyond the strain range, a large area of bone will be fractured. This is consistent with the results of this study; when the speed reached a certain limit, fracture occurred. In this study, cracks first appeared on the outer and upper sides. After the bone fractured on the outer and upper sides of the femoral neck, the integrity of the component was damaged, and the cracks grew rapidly. This was because compared with the inner and lower side of the femoral neck, the bone density on the upper and outer sides of the femoral neck was lower, and the shearing effect when resisting lateral falls was weak [33]. In addition, fracture mechanics studies have shown that the strain rate of cancellous bone and cortical bone increased with the increase of the impact speed, and the fracture toughness and compressive strength of the bone also increased [34,35]. They also pointed out that the effect of impact speed on fracture is actually macroscopic performance after the impact speed changes various material properties of bone. Zuo et al. [36] pointed out that in geotechnical engineering, the failure of linear and isotropic rock samples increased with the increase of the impact speed, the crack tended to be more complex, and greater energy was released during failure. Similarly, the fracture model in this study also showed a trend that the fracture line became complex and rough as the impact speed increased. In addition, the fracture load of the model gradually decreased with the increase of the impact speed, which was similar to the literature [34,35].

When the fall postures were different, it was found that with the increase of α angle, the maximum principal stress and the maximum principal strain of the femur also increased. Falling forward was less likely to cause composite proximal femoral fracture compared with falling backward. The fracture load decreased by 27.37% as α angle changed from 15° to 135°. Ford and Kryak et al. [37,38] also reached conclusions that were basically consistent with this study, which pointed out that a 26% reduction in load capacity was equivalent to the result of bone mineral density loss of 25 years after the age of 65. The structural capacity of the proximal femur, like any other structure, is determined by the applied loads, which can vary depending on the direction of impact during a fall. Our findings demonstrate the independent contribution of fall mechanics to hip fracture risk by identifying a fall aspect (the direction of impact) that is an important determinant of fall severity. This is also consistent with the results of Fu et al. [39]. It could be seen that the cortical thicknesses had great effects on the fracture types and mechanical responses. As the cortical bone became thinner, the fracture line gradually moved from the composite femoral neck to the intertrochanter, resulting in different types of composite proximal femoral fractures. It could be seen that the thickness of cortical bone was an important factor affecting the types of composite proximal femoral fractures. Henning et al. [40] reached similar conclusions. The study also pointed out that the thinning of cortical bone with aging led to a gradual decrease in the strength and stiffness of the proximal femur, which was the main factor that caused the elderly to be more prone to fracture [41]. Cortical bone is the main load-bearing region of bone. In the femoral neck, cortical bone contributes higher than cancellous bone to bone strength by 4.6–17.3% [42], but cancellous bone plays an important role in resisting bending deformation. It was pointed out that both cortical thickness and CTI are significant risk factors for proximal femoral fracture [43,44]. Thus, it is essential to understand the relative contribution of cortical thickness for the treatment and prevention of fractures in clinics.

The changes of fall speeds and postures can cause great energy absorption, and different cortical thicknesses in the proximal femur can cause different structural changes, which might be one of the main factors leading to proximal femoral fracture [45]. Ertas et al. [46] simulated 10 heterogeneous samples using an empirically validated creep strain accumulation model to determine the relationship between steady-state creep rate, applied load, and microstructure. They stated that impact speed is one of the factors that contribute to the mechanism of viscoelastic mechanical properties of human cortical bone, and that it is important for predicting bone response to creep and fatigue loading. They also investigated the creep behavior of porcine cancellous bone, finding strong relationships between applied stress and both time-to-failure and steady-state creep rate, which is consistent with our findings [47]. Because XFEM could not simulate sudden changes in the proximal femoral speed and acceleration during the falling process, the results were therefore different from the results of mechanical tests. In this study, IED-based FEA was found to be effective to simulate different fall conditions, and the crack propagation path and fracture load with time during the fall were analyzed. The model could also be used to predict proximal femoral fractures caused by other conditions such as dropping from high places or impacts in various situations. Therefore, it could help to understand the biomechanical mechanism of proximal femoral fracture by investigating the crack propagation path and the final damage state. The model could also be assembled with internal fixation devices to simulate proximal femur internal fixation and simulation verification of mechanical properties of existing metal implants to guide the improvement of internal fixation devices and intraoperative fixation decisions [48,49].

However, this study still has some limitations and needs to be further improved. Only the composite femur was used to establish a numerical model in which simply two different material properties of cortical bone and cancellous bone were assigned. Although MacLeod et al. [50] showed that composite femurs could replace human bones to do mechanical tests, the composite femurs cannot completely simulate human femurs, since the materials and morphology of human femurs are more complex. In the future, material properties that are closer to the real human femur should be used. At the same time, only the femur was modeled in this study, and the muscles, fascias, and ligaments were not simulated. In the future, mechanical tests could be performed to obtain the loads of muscles, fascias, and ligaments on bones [51], and a new FE model of proximal femur should be established. The results obtained would be closer to reality. Although this study did not carry out direct experimental verification of all models, the crack propagation path and the predicted fracture load in the simulated standing state were highly similar to those obtained by mechanical tests. Thus, the computational model could be considered effective and used to analyze other conditions. It could be further used to simulate bone implants, fractures caused by pedestrian impacts with cars, and more types of fractures. In addition, it could provide a more theoretical basis for fracture research.

## 5. Conclusions

In this study, IED-based FEA was developed and compared with XFEM. Moreover, the effects of different impact speeds, fall postures, and cortical thicknesses on fracture types and mechanical responses were investigated by IED-based FEA. IED-based FEA was found to better simulate the occurrence and development of composite proximal femoral fracture than XFEM with comparisons of mechanical tests. It could well predict the effects of different fall conditions on fracture types and mechanical responses. When the speed was faster, the time of fracture was shorter, and the crack line moved down significantly. When the α angle changed from 15° to 135°, the fracture load decreased by 27.37%, indicating that falling forward was less likely to cause proximal femoral fracture compared with falling backward. Moreover, the model with thin cortical bone was prone to fracture, and when the entire model was cancellous bone, the femoral head was fractured. The study conducted a comprehensive theoretical analysis of proximal femoral fractures, which may provide sufficient theoretical support in the development of a prevention methodology of femoral fractures. Through this technique, it is possible to simulate long fracture paths, which is important when fracture morphology is studied, since different fracture morphologies must be treated with distinct surgical treatments.

## Figures and Tables

**Figure 1 materials-15-02878-f001:**
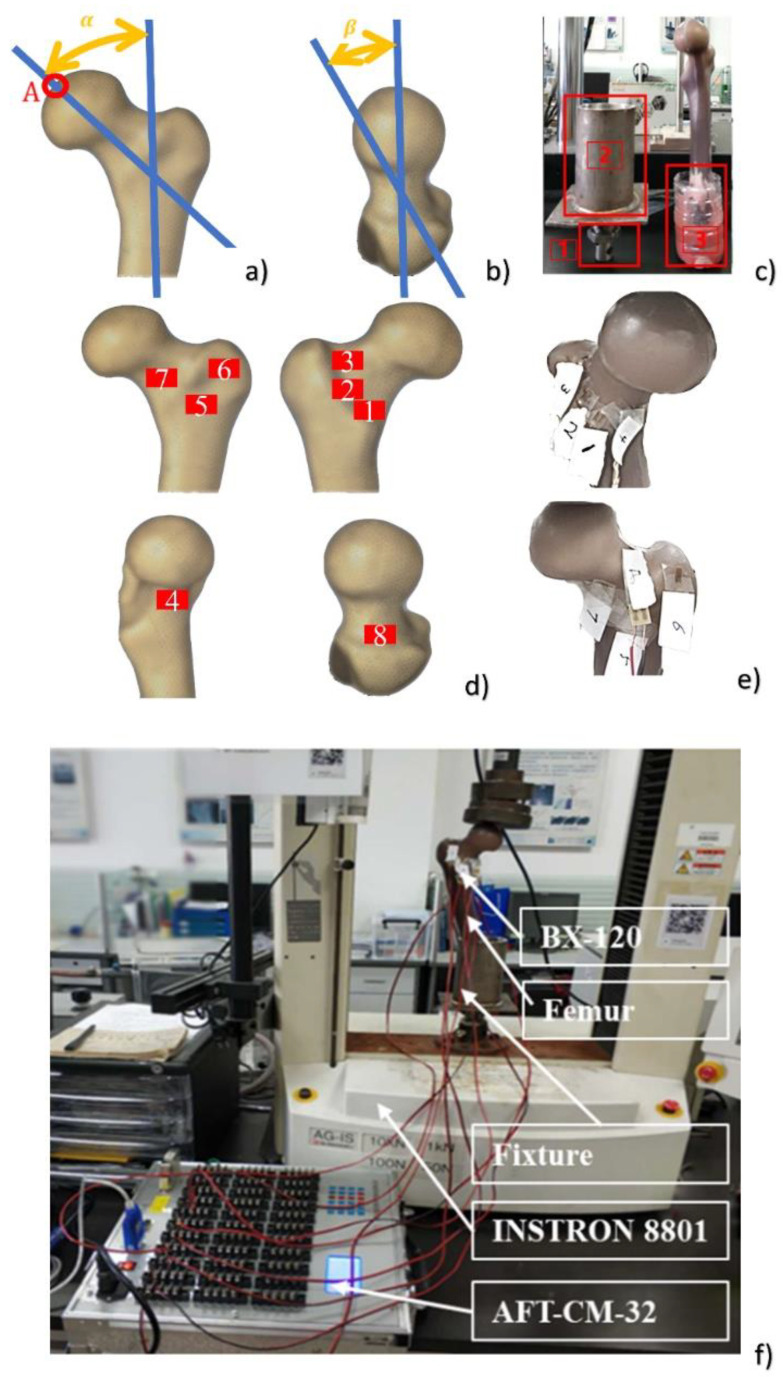
The experimental layout of mechanical tests: (**a**) angle α (0–135°) with reference to the long axis of the femur in the frontal plane; (**b**) angle β (0–45°) with reference to the femoral neck axis in the horizontal plane; (**c**) fixation method and device of the composite femur; (**d**) position of strain gages on the femur in the numerical simulation; (**e**) position of strain gages on the femur in mechanical tests; (**f**) the experimental device of composite femur compression. Point A was the central position of the spherical region surface with a 35 mm diameter on the femoral head.

**Figure 2 materials-15-02878-f002:**
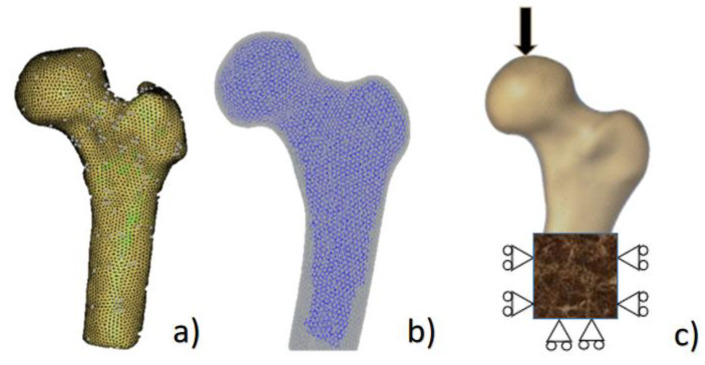
The composite proximal femur with 3D tetrahedral mesh, materials, and loading condition. (**a**) The meshed composite proximal femur; (**b**) the composite proximal femur with two materials: the outer transparent part is cortical bone, and the inner opaque part is cancellous bone; (**c**) the boundary condition and loading condition.

**Figure 3 materials-15-02878-f003:**
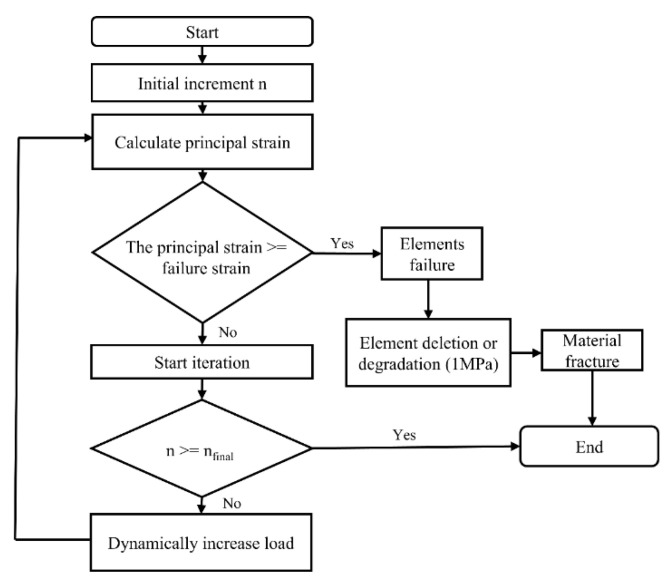
Scheme of the successive analysis programmed through IED script.

**Figure 4 materials-15-02878-f004:**
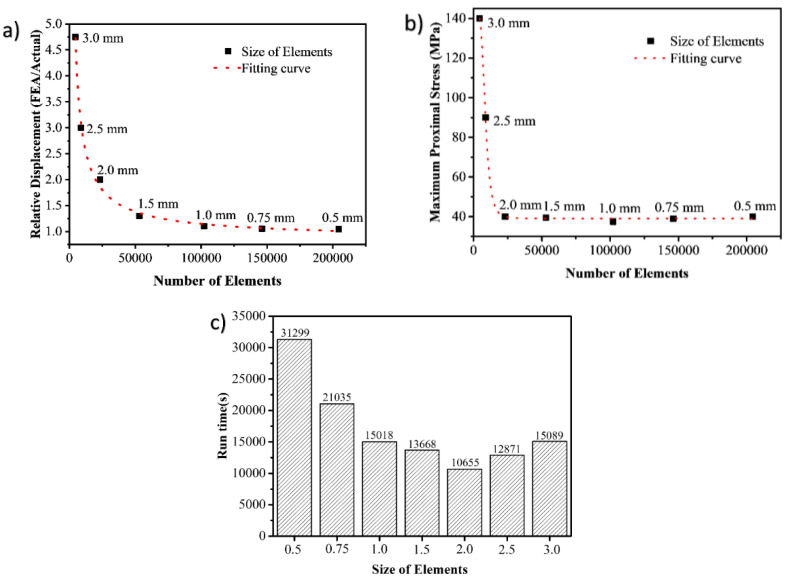
The effects of different mesh sizes on the FE results: (**a**) relative displacement (FEA/Actual) means the ratio of simulated displacement and experimental displacement of point A (Point A is shown in Figure 1); (**b**) the maximum principal stress of the composite proximal femur; (**c**) running time of the FE results.

**Figure 6 materials-15-02878-f006:**
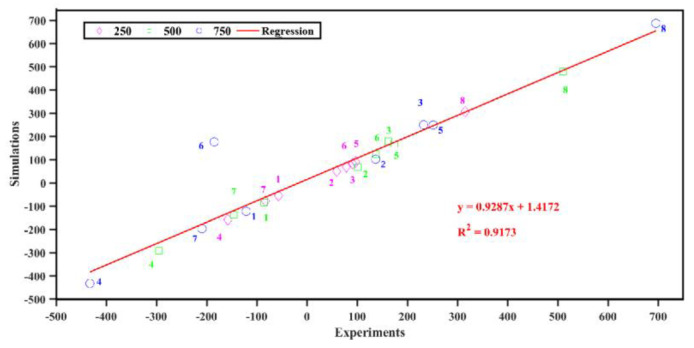
Comparison between the numerical model and experiment. The numbers next to the markers indicate the gage position number given in Figure 1d.

**Figure 7 materials-15-02878-f007:**
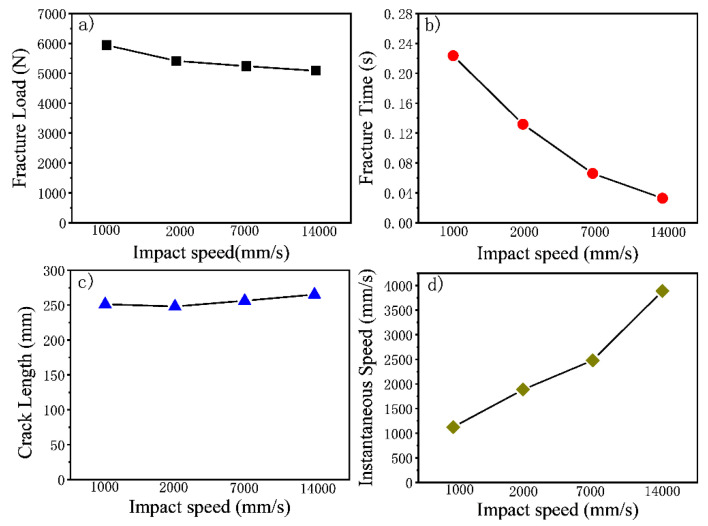
The effects of impact speed on the mechanical responses of composite proximal femoral fractures: (**a**) fracture load; (**b**) fracture time; (**c**) crack length; (**d**) instantaneous speed.

**Figure 8 materials-15-02878-f008:**
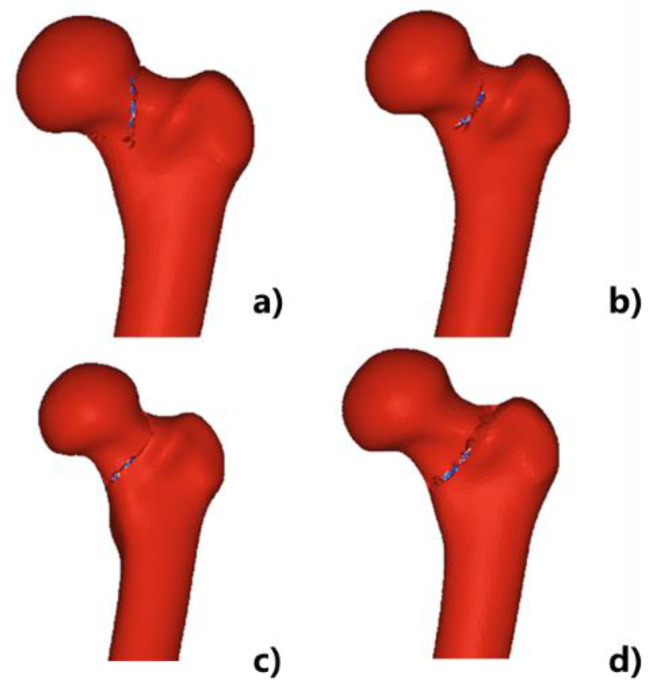
Simulated fracture crack propagation paths of composite proximal femur with different impact speeds: (**a**) 1000 mm/s; (**b**) 2000 mm/s; (**c**) 7000 mm/s; (**d**) 14,000 mm/s.

**Figure 9 materials-15-02878-f009:**
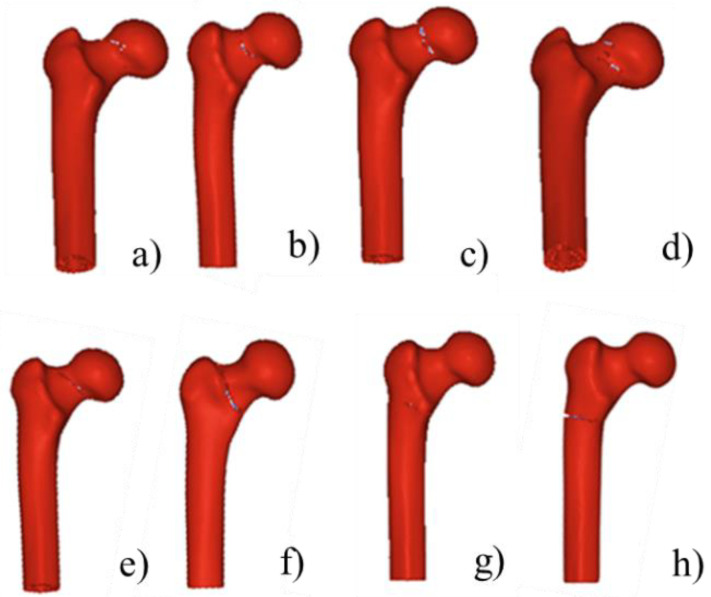
Eight typical fracture crack propagation paths obtained by numerical simulation, including femoral neck fracture and intertrochanteric fracture, and the fall postures: (**a**) α = 20°, β = 0°; (**b**) α = 75°, β = 0°; (**c**) α = 0°, β = 15°; (**d**) α = 55°, β = 0°; (**e**) α = 0°, β =30°; (**f**) α = 40°, β = 30°; (**g**) α = 100°, β = 30°; (**h**) α = 80°, β = 45°.

**Figure 10 materials-15-02878-f010:**
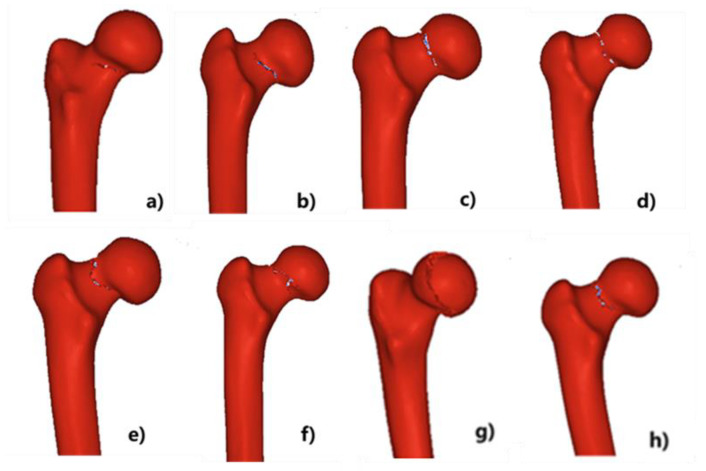
Crack propagation paths of the composite femur models with different thicknesses, and the percentages of cortical bone mass/bone mass: (**a**) 99%; (**b**) 80%; (**c**) 60%; (**d**) 40%; (**e**) 20%; (**f**) 10%; (**g**) 1%. (**h**) Hollow bone model.

**Table 2 materials-15-02878-t002:** When the angle β was constant, the *p*-value between the angle α and the maximum principal stress, the maximum principal strain, and fracture time.

Angle	The Maximum Principal Stress	The Maximum Principal Strain	Fracture Time
**β** = 0°	0.2315	0.1259	0.2013
**β** = 15°	**0.075 ***	**0.0083 ***	0.2312
**β** = 30°	0.1876	0.1974	0.2908
**β** = 45°	0.1565	0.2215	0.2561

* The data of significant correlation.

**Table 3 materials-15-02878-t003:** Effects of different cortical thicknesses on the mechanical responses.

Cortical Bone Mass/Bone Mass	Cortical Bone		Cancellous Bone		Fracture Load (N)	Maximum Principal Stress (Pa)	Maximum Principal Strain
	Elements	Nodes	Elements	Nodes			
99%	80,693	14,748	-	-	7329	1.406 × 10^8^	0.994 × 10^−2^
80%	77,649	13,742	3044	1006	7100	2.945 × 10^8^	1.209 × 10^−2^
60%	68,650	11,742	12,043	3016	6550	3.197 × 10^8^	1.234 × 10^−2^
40%	54,425	9391	26,268	5391	6345	3.557 × 10^8^	1.276 × 10^−2^
20%	48,918	8773	31,775	5945	5701	3.621 × 10^8^	1.278 × 10^−2^
10%	33,918	5773	46,775	8975	5039	3.856 × 10^8^	1.291 × 10^−2^
1%	-	-	70,693	14,748	4575	3.979 × 10^8^	1.301 × 10^−2^
Hollow	54,425	9391	-	-	6243	3.457 × 10^8^	1.076 × 10^−2^

## Data Availability

The study did not report any data.

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
