# Peer review of "Incremental Element Deletion-Based Finite Element Analysis of the Effects of Impact Speeds, Fall Postures, and Cortical Thicknesses on Femur Fracture"

_materials, 2022, doi:10.3390/ma15082878_

Round 1

Reviewer 1 Report

Dear authors,

the study is properly designed with limitations highlighted (i.e. only SAW bones) and the results provide an insight into bone fracture mechanics without complicated methods like XFEM or Phasefields. The reviewer believes the study is ready to be considered for publishing.

Author Response

MaterialsRef: materials-1628806Title: Effects of Impact Speeds, Fall Postures and Cortical Thicknesses on Femur Fracture by Incremental Element Deletion Based Finite Element AnalysisDear editor and reviewers:

Thank you very much for your letter and advice.

We appreciate editor and reviewers very much for their positive and constructive comments and suggestions on our manuscript entitled “Effects of Impact Speeds, Fall Postures and Cortical Thicknesses on Femur Fracture by Incremental Element Deletion Based Finite Element Analysis” (Manuscript ID: materials-1628806). We have studied the comments of both the editor and reviewers carefully and tried our best to revise the paper accordingly. Point-to-point replies are included as below.

We would like to re-submit the revised manuscript for your consideration. I hope that the revision is acceptable for publication in your journal. Looking forward to hearing from you soon.

With best regards,

Yours sincerely,

Weiqiang Liu

************************************************************

List of replies

Dear editor and reviewers:

We would like to express our sincere thanks to you for the constructive and positive comments on our manuscript entitled “Effects of Impact Speeds, Fall Postures and Cortical Thicknesses on Femur Fracture by Incremental Element Deletion Based Finite Element Analysis” (Manuscript ID: materials-1628806). The revisions have been made accordingly and highlighted in the revised manuscript with tracked changes.

Reviewers' comments:

Reviewer #1:

Point 1: Dear authors,

the study is properly designed with limitations highlighted (i. e. only SAW bones) and the results provide an insight into bone fracture mechanics without complicated methods like XFEM or Phase fields. The reviewer believes the study is ready to be considered for publishing.

Response 1: Thank you for your careful review and valuable suggestions. We would like to express our sincere thanks again to you for the constructive and positive comments on our manuscript entitled “Effects of Impact Speeds, Fall Postures and Cortical Thicknesses on Femur Fracture by Incremental Element Deletion Based Finite Element Analysis” (Manuscript ID: materials-1628806). We have optimized the article again this time, the grammar errors, sentence structure and sentence fragments, etc. had been revised in revised manuscript.

Reviewer 2 Report

In the entitled paper ‘Effects of Impact Speeds, Fall Postures and Cortical Thicknesses on Femur Fracture by Incremental Element Deletion Based Finite Element Analysis’, the authors developed an IED technique, based on FEA and compared it with XFEM and several parameters were investigated. The paper needs some improvements to be accepted for publication. Please! See the following comments:

- The abstract needs to be improved to show the originality of this paper.

- In Line 35, the authors mentioned about the increase in 2050, I think there is no meaning to say this sentence because it difficult to predict due to a lot of changes in human's life conditions, otherwise, the authors should argue.

- Line 64, ‘These are difficult to reflect the dynamic expansion process of crack’. This sentence should be reformulated.

- Line 72, How can the authors use the adjective ‘reliable theoretical guidance’. The word reliability is generally related to statistical studies.

- It is better to reduce the caption of Figure 1 and to move some sentences to the text.

- The authors ignored the anisotropy during their study. How do they justify this ignorance?

- The fractures in Figures 6, 8, 9 and 10 are not clear. It is better to find a way to clarify them.

- Some figures containing stress and stain should be added to show more the significant results.

- During the study, the authors considered the thickness of the cortical bone but not the density. In fact, we should not ignore the density effect when dealing with this kind of problems. How do they justify this ignorance?

Author Response

MaterialsRef: materials-1628806Title: Effects of Impact Speeds, Fall Postures and Cortical Thicknesses on Femur Fracture by Incremental Element Deletion Based Finite Element AnalysisDear editor and reviewers:

Thank you very much for your letter and advice.

We appreciate editor and reviewers very much for their positive and constructive comments and suggestions on our manuscript entitled “Effects of Impact Speeds, Fall Postures and Cortical Thicknesses on Femur Fracture by Incremental Element Deletion Based Finite Element Analysis” (Manuscript ID: materials-1628806). We have studied the comments of both the editor and reviewers carefully and tried our best to revise the paper accordingly. Point-to-point replies are included as below.

We would like to re-submit the revised manuscript for your consideration. I hope that the revision is acceptable for publication in your journal. Looking forward to hearing from you soon.

With best regards,

Yours sincerely,

Weiqiang Liu

************************************************************

List of replies

Dear editor and reviewers:

We would like to express our sincere thanks to you for the constructive and positive comments on our manuscript entitled “Effects of Impact Speeds, Fall Postures and Cortical Thicknesses on Femur Fracture by Incremental Element Deletion Based Finite Element Analysis” (Manuscript ID: materials-1628806). The revisions have been made accordingly and highlighted in the revised manuscript with tracked changes.

Reviewers' comments:

Reviewer #2:

Point 1: In the entitled paper ‘Effects of Impact Speeds, Fall Postures and Cortical Thicknesses on Femur Fracture by Incremental Element Deletion Based Finite Element Analysis’, the authors developed an IED technique, based on FEA and compared it with XFEM and several parameters were investigated. The paper needs some improvements to be accepted for publication. Please! See the following comments:

Response 1: Thank you for your careful review and valuable suggestions. We appreciate editor and reviewers very much again for your positive and constructive comments and suggestions on our manuscript entitled “Effects of Impact Speeds, Fall Postures and Cortical Thicknesses on Femur Fracture by Incremental Element Deletion Based Finite Element Analysis” (Manuscript ID: materials-1628806). We have studied the comments of both the editor and reviewers carefully and tried our best to revise the paper accordingly. Point-to-point replies are included as below.

Point 2: The abstract needs to be improved to show the originality of this paper.

Response 2: Thank you for your careful review and valuable suggestions. To show the originality of this paper, the abstract of this paper has been refactored as followed. (Lines 20-31).

The proximal femur's numerical simulation could give an effective method for predicting the risk of femoral fracture. However, the majority of existing numerical simulations are static simulations, which do not correctly capture the dynamic properties of bone fractures. On the basis of femoral fracture analysis, a dynamic simulation using incremental element deletion (IED) based finite element analysis (FEA) was developed and compared to XFEM in this study. Mechanical tests were also used to assess it. Different impact speeds, fall postures, and cortical thicknesses were also studied for their implications on fracture types and mechanical responses. The time it took for the crack to shatter was shorter when the speed was higher, and the crack line slid down significantly. The time it took for the crack to shatter was shorter when the speed was higher, and the crack line slid down significantly. The fracture load fell by 27.37 percent when the angle was altered from 15° to 135°, indicating that falling forward was less likely to cause proximal femoral fracture than falling backward. Furthermore, the model with scant cortical bone was susceptible to fracture. This research established a theoretical foundation and mechanism for forecasting the risk of proximal femoral fracture in the elderly.

Point 3: In Line 35, the authors mentioned about the increase in 2050, I think there is no meaning to say this sentence because it difficult to predict due to a lot of changes in human's life conditions, otherwise, the authors should argue.

Response 3: Thank you for your careful review and valuable suggestions. As the reviewer said, there is no meaning to say about the increase in 2050, so this sentence was deleted as followed. (Lines 36-38).

The number of patients with proximal femoral fractures continues to grow every year, there is currently one proximal femoral fracture every 3 s in the world [2].

Point 4: Line 64, ‘These are difficult to reflect the dynamic expansion process of crack’. This sentence should be reformulated.

Response 4: Thank you for your careful review and valuable suggestions. The sentence ‘These are difficult to reflect the dynamic expansion process of crack’ was changed to ‘The extant numerical simulations are mostly static ones, which are difficult to represent the dynamic crack expansion process [8,14].’, as followed. (Lines 66-67).

The extant numerical simulations are mostly static ones, which are difficult to represent the dynamic crack expansion process [8,14].

Point 5: Line 72, How can the authors use the adjective ‘reliable theoretical guidance’. The word reliability is generally related to statistical studies.

Response 5: Thank you for your careful review and valuable suggestions. As the reviewer said, the word reliability is generally related to statistical studies. However, in this study, this sentence expresses the importance of this research and has nothing to do with statistics. As a result, the sentence has been restructured as followed. (Lines 72-75).

It potentially provided a trust worthy method for indicating the proximal femoral fracture risk in the elderly, which can serve as a theoretical foundation for clinical orthopedics and surgical procedure evaluation.

Point 6: It is better to reduce the caption of Figure 1 and to move some sentences to the text.

Response 6: Thank you for your careful review and valuable suggestions. The caption of Figure 1 was reduced and some sentences was moved to the text as followed. (Lines 111-116).

Figure 1. The experimental layout of mechanical tests. a) The angle α (0°-135°) with reference to the long axis of the femur in the frontal plane; b) The angle β (0°-45°) with reference to the femoral neck axis in the horizontal plane; c) Fixation method and device of the composite femur; d) Position of strain gages on the femur in the numerical simulation; e) Position of strain gages on the femur in mechanical tests; f) The experimental device of composite femur compression.

Point 7: The authors ignored the anisotropy during their study. How do they justify this ignorance?

Response 7: Thank you for your careful review and valuable suggestions. The assignment of mechanical parameters to the finite element model is an important step in mechanical simulation and serves as the foundation for realizing the finite element model's mechanical properties. At the moment, the most commonly used mechanical parameter assignment methods are: (1) uniformly assigning material properties; and (2) assigning material parameters based on gray value. (3) manually setting each part's thickness and assigning material parameters separately (4) In general, each part of the specimen's measured parameters is assigned. This is the closest method to the real thing. However, the material of each part of the skeleton in the human body is not uniform, and the mechanical properties are also quite different, making actual operation extremely difficult. Some studies divide the elastic modulus of the bone into 256 grades and correct the CT value corresponding to the material properties of each unit in order to be closer to the bone elastic modulus parameters of each part of the human specimen during bone modeling. The calculation of the conversion relationship between the density value and the elastic modulus is especially important for patients with changes in bone density, but it is a more complicated operation. As a result, according to some literature, the proximal femur model can be simplified as an isotropic linear material under static load, and the proximal femur is mechanically assigned based on a continuous, homogeneous, and isotropic linear elastic material. The mechanical model of the femoral end can be very close to the actual specimen, with a high degree of anatomical shape similarity and mechanical simulation. To some extent, it can replace the artificial composite proximal femur specimen for mechanical testing and analysis.

Point 8: The fractures in Figures 6, 8, 9 and 10 are not clear. It is better to find a way to clarify them.

Response 8: Thank you for your careful review and valuable suggestions. We attempted to show the results with a partial zoom (as shown in Figure1), but the location and length of the fracture were difficult to distinguish. In contrast to the partial zoom representation, the overall fracture map representation is more appropriate, as it can be more focused. So, in the end, we relied on the overall graph depicted in Figure 8. The other figures are there for the same reason.

Figure 1. The initial form of expression of Figure 8.

Point 9: Some figures containing stress and stain should be added to show more the significant results.

Response 9: Thank you for your careful review and valuable suggestions. As shown in the Figures 6, 8, 9 and 10, are primarily intended to represent the shape and propagation path of the crack. It has been noted in the paper if it is necessary to express the stress or strain of other meanings.

Point 10: During the study, the authors considered the thickness of the cortical bone but not the density. In fact, we should not ignore the density effect when dealing with this kind of problems. How do they justify this ignorance?

Response 10: Thank you for your careful review and valuable suggestions. In this study, a finite element model of the proximal femur including cortical bone and cancellous bone was reconstructed, and different thicknesses of cortical bone were set. It was found that as the cortical thickness decreased, the fracture line gradually moved down from the femoral neck to the intertrochanteric, and the femoral cortical bone thickness was Important factors affecting the type of hip fracture. The reason why this study only includes cortical bone and cancellous bone is that the artificial composite femur can be better simulated. part. It is also by setting different thicknesses of cancellous bone and cortical bone to simulate artificial composite femurs of different ages. Secondly, the method of assigning the density of the two materials in this study is also simplified in this method in most studies, and the feasibility of this method is proved.

Reviewer 3 Report

First and foremost, the topic is so important that I am hopeful that this study will result in some useful improvements in predicting femoral fracture risk, particularly in the elderly. The authors must, however, consider the following comments/corrections before publishing. This is critical from an ethical standpoint as well as the study's readability. The following is a list of the above-mentioned comments and corrections:

  • The title and the majority of the abstract were copied verbatim from a website, which is unethical and unacceptable. To prevent similarities with existing publications, the title and abstract could be altered to the following.

Title suggestion:Incremental Element Deletion Based Finite Element Analysis of the Effects of Impact Speeds, Fall Postures, and Cortical Thicknesses on Femur Fracture

Abstract suggestion:The proximal femur's numerical simulation could give an effective method for predicting the risk of femoral fracture. However, the majority of existing numerical simulations are static simulations, which do not correctly capture the dynamic properties of bone fractures. On the basis of femoral fracture analysis, a dynamic simulation using incremental element deletion (IED) based finite element analysis (FEA) was developed and compared to XFEM in this study. Mechanical tests were also used to assess it. Different impact speeds, fall postures, and cortical thicknesses were also studied for their implications on fracture types and mechanical responses. The time it took for the crack to shatter was shorter when the speed was higher, and the crack line slid down significantly. The time it took for the crack to shatter was shorter when the speed was higher, and the crack line slid down significantly. The fracture load fell by 27.37 percent when the angle was altered from 15° to 135°, indicating that falling forward was less likely to cause proximal femoral fracture than falling backward. Furthermore, the model with scant cortical bone was susceptible to fracture. This research established a theoretical foundation and mechanism for forecasting the risk of proximal femoral fracture in the elderly.

  • I have some suggestions for sentences that were taken directly from websites or documents. It is unethical to use idies from sources without making any alterations, even if references are used for the aforementioned lines.

Suggestion for the sentences (page 2, lines 50-55):Bone fracture may now be studied at both the micro and macro scales because to advances in computer modeling. With a correlation of nearly 90%, linear finite element models have been effectively used to the prediction of the elastic response and fracture load of a human femur [8]. Despite efforts to simulate human femur behavior, fracture patterns have only been modelled using numerical methodologies on a few occasions. Some studies have focused on proximal fracture simulation, with the majority of them yielding short fracture pathways [9-13].

Suggestion for the sentences (page 2, lines 77-86): Instead of saying “The composite femur has been commonly used in the literature as a simulant of real bone. It is important to emphasise that this kind of specimens is designed to simulate the biomechanical properties of young and healthy femurs [15,16]. These similarities were tested by means of axial compression, bending and torsion tests through the measurement of the corresponding stiffness and ultimate failure strength [15,16]. The use of artificial bone provides advantages for model validation avoiding the variability of properties inherent to biological tissues. Composite bones are useful to develop controlled analysis, due to their homogeneous properties in two distinct zones, smoothed surface and low variability between specimens [15]. The failure modes of these composite models are close to published findings for human bones [16].” I suggest the following wording: “In the literature, the composite femur has been widely employed as a substitute for actual bone. It's vital to note that these specimens are intended to mimic the biomechanical qualities of young, healthy femurs [15,16]. Axial compression, bending, and torsion tests were used to assess these commonalities, with the related stiffness and ultimate failure strength measured [15,16]. Artificial bone has advantages for model validation since it eliminates the variety of properties found in biological tissues. Because of their uniform qualities in two separate zones, smoothed surface, and minimal variability between specimens, composite bones are excellent for developing controlled analysis [15]. These composite models' failure patterns are similar to those reported for human bones [16].

Suggestion for the sentences (page 4, lines 120-123): Instead of saying “The loads were applied to the femur in the stance loading direction (Fig. 2a). This position is the common configuration analysed in the literature because it corresponds to the physiological scheme of a standingup human position[19].” I suggest the following wording: “The loads were applied to the femur in the direction of stance loading (Fig. 2a). Because it matches to the physiological plan of a standing-up human position, this is the most commonly studied configuration in the literature.

Suggestion for the sentences (page 4, lines 126-129): Instead of saying “Load was increased up to the different values considered (250 N, 500 N and 750 N with an actuator speed of 0.3 mm/s in order to keep quasi-static conditions). The femur linear elastic behaviour was verified through this test, as reported by other author [20].” I suggest the following wording: “To maintain quasi-static conditions, the load was increased up to the different values evaluated (250 N, 500 N, and 750 N with an actuator speed of 0.3 mm/s). This test confirmed the linear elastic behavior of the femur, as described by another author [20].

Suggestion for the sentences (pages 4-5, lines 155-158): Instead of saying “The boundary conditions and the loading state are set as close as possible to the conditions imposed during the experiments. The distal part of the diaphysis is constrained to the testing rig simulating the embedment in surgical cement and the load was applied in the proper direction similar to the configuration used in the experiments.” I suggest the following wording: “The boundary conditions and loading state are set to be as close to the experimental conditions as possible. The distal part of the diaphysis is restrained to the testing rig, which simulates surgical cement embedment, and the force is imparted in the right direction, as in the tests.

Suggestion for the sentences (page 7, lines 240-244): Instead of saying “The developed models reproduce the mechanical behaviour of the loading of composite femurs defined in the experiments. In the elastic regime, femurs were instrumented with 8 strain gages. Since the three measurements were recorded at 3 different loading steps (250 N, 500 N, 750 N), a total of 24 strain datas were registered for each femur to validate the corresponding numerical model.” I suggest the following wording: “The generated models accurately represent the mechanical behavior of composite femur loading as seen in the experiments. Femurs were fitted with eight strain gages in the elastic regime. Because the three measurements were taken at three different loading levels (250 N, 500 N, and 750 N), each femur had a total of 24 strain data points to validate the numerical model.

Suggestion for the sentences (page 8, lines 254-260): Instead of saying “Regarding the predicted fracture location and early stages of crack growth shown in Fig. 6a-b, it can be seen that it is similar to the failure region observed in Fig. 6c. Moreover, the numerical results showed the same trends obtained by other authors also with composite femurs [27]. Concerning the fracture loading, the experimental test yielded a maximum load of 6573±10 N. The fracture test occurred at the neck, as it is also found in the works by other author [28]. Our numerical simulations also predict the failure at this zone; see the fracture path calculated for the stance loading case in Fig. 6d..” I suggest the following wording: “It can be seen in Fig. 6a-b that the projected fracture location and early phases of crack propagation are identical to the failure zone seen in Fig. 6c. Furthermore, the numerical results confirmed the same tendencies found by other researchers using composite femurs [27]. The experimental test revealed a maximum load of 6573±10 N in terms of fracture loading. The fracture test was performed on the neck, as described by other authors [28]. Our numerical simulations similarly anticipate failure in this zone; see Fig. 6d for the projected fracture route for the stance loading case.

  • The first paragraph (page 12, lines 371-379) in the discussion section was copied verbatim from a website and needs to be changed with the authors' own words. The following is an example of a suggestion: “IED-based FEA results in longer fracture trajectories than XFEM. Due to convergence issues, XFEM produced unsatisfactory results for long crack trajectories. As a result, pathways generated using an IED-based FEA showed good convergence behavior, resulting in extended trajectories. Because each increment of the fracture growth represents a fresh simulation, this technique avoids convergence issues. When comparing the two procedures, element elimination has more issues due to the presence of distorted elements, which might cause the numerical process to slow down. As a result, the IED-based FEA technique produces the best results in terms of convergence and fracture path length, and it can be applied to a variety of loading scenarios.
  • The usage of a 3D tetrahedral mesh in FE analysis is discussed on page 4, lines 146-154, but no element specifics are stated. As a result, please include any element details used in the FE analysis.
  • The regression analysis findings are a little dubious. As a result, I recommend including values for the mean, standard deviation, and COV in the experimental results. Furthermore, the confidence interval (e.g., for 95% and 99%) values of the experiments will improve the results' reliability as well as the paper's quality.
  • Some minor typos should be corrected.

For instance:

-Page 1, line 43, replace “on” by “in”

-Page 2, line 50, replace “modelling” by “modeling” because the spelling of modelling (with double l) is a non-American variant. For consistency, consider replacing it with the American English spelling.

-Page 2, line 64, replace the second “simulations” by “ones” or delete that word.

-Page 2, line 78, replace “specimens” by “specimen” (without “s”)

-Page 4, line 126, Before “load” please use “The”.

-Page 4, line 129, replace “author” by “authors”

-Page 4, line 137, add “the” before “simulation”

-Page 4, line 146, add “the” before “element”

-Page 4, line 146, the “the” in front of the “both” should be deleted.

-Page 5, line 160, Before “loading” please use “The”.

-Page 5, line 162, replace “author” by “authors”

-Page 5, line 171, use comma (,) before “and”

-Page 5, lines 171 and 172, replace “An initial increment n of load was set….” by “An initial increment "n" of the load was set…”

-Page 5, line 177, Before “principal” please use “the”.

-Page 5, line 178, the “the” in front of the “Young’s modulus” should be deleted.

-Page 6, line 187, Before “XFEM” please use “the”.

-Page 6, line 191, use comma (,) before “and”

-Page 6, line 194, the comma (,) in front of the “and” (second one) should be deleted.

-Page 6, line 220, add “was” before “fractured”

-Page 7, line 243, replace “datas” by “data”.

-Page 8, line 251, Before “relative” please use “The”.

-Page 8, line 252, add “the” before “fracture”

-Page 8, line 259, replace “author” by “authors”

-Page 9, line 279, replace “of” by “in”

-Page 9, line 283, replace “thes” by “the”

-Page 9, line 284, please add “that” before “with”

-Page 9, line 285, please add “the” before “human”

-Page 9, line 290, the “the” in front of the “four” should be deleted.

-Page 10, line 301, use comma (,) before first “and”

-Page 10, line 302, use comma (,) before “and”

-Page 10, line 340, delete one of the “the”

-Page 10, line 360, replace “resulted” by “resulting”

-Page 12, line 374, the comma (,) in front of the “because” should be deleted.

-Page 12, line 384, replace “literatures” by “literature”

-Page 12, line 384, please add “that” before “when”

-Page 12, line 387, please add “s” after “side”

-By the way, the reference numbers are written next to the final words of each phrase. Before the reference numbers, a gap should be left.

  • The conclusion should be written in bullet points with only the most important findings.
  • The following studies must be evaluated and cited because the authors have used some information from them directly or indirectly, or because there are relevant research that the authors have not discussed or cited. The following are the most important studies:
  • “The effect of load direction on the structure capacity of human proximal femur during falling” in Applied Mechanics and Materials, vol. 137, pp. 7-11, 2011
  • “Simulation of creep in non-homogenous samples of human cortical bone” in Computer Methods in Biomechanics and Biomedical Engineering, vol. 15, pp. 1121-1128, 2012
  • “Creep simulation of a micro-CT based finite element model of porcine cancellous bone” in Summer Engineering Conference, 54587, pp. 285-286, 2011
  • Finally, while the work is well-written in general, it does contain some grammatical and typographical problems. Before resubmitting the manuscript, it is suggested that the authors reread it again.

Author Response

MaterialsRef: materials-1628806Title: Effects of Impact Speeds, Fall Postures and Cortical Thicknesses on Femur Fracture by Incremental Element Deletion Based Finite Element AnalysisDear editor and reviewers:

Thank you very much for your letter and advice.

We appreciate editor and reviewers very much for their positive and constructive comments and suggestions on our manuscript entitled “Effects of Impact Speeds, Fall Postures and Cortical Thicknesses on Femur Fracture by Incremental Element Deletion Based Finite Element Analysis” (Manuscript ID: materials-1628806). We have studied the comments of both the editor and reviewers carefully and tried our best to revise the paper accordingly. Point-to-point replies are included as below.

We would like to re-submit the revised manuscript for your consideration. I hope that the revision is acceptable for publication in your journal. Looking forward to hearing from you soon.

With best regards,

Yours sincerely,

Weiqiang Liu

************************************************************

List of replies

Dear editor and reviewers:

We would like to express our sincere thanks to you for the constructive and positive comments on our manuscript entitled “Effects of Impact Speeds, Fall Postures and Cortical Thicknesses on Femur Fracture by Incremental Element Deletion Based Finite Element Analysis” (Manuscript ID: materials-1628806). The revisions have been made accordingly and highlighted in the revised manuscript with tracked changes.

Reviewers' comments:

Reviewer #3:

Point 1: First and foremost, the topic is so important that I am hopeful that this study will result in some useful improvements in predicting femoral fracture risk, particularly in the elderly. The authors must, however, consider the following comments/corrections before publishing. This is critical from an ethical standpoint as well as the study's readability. The following is a list of the above-mentioned comments and corrections:

Response 1: Thank you for your careful review and valuable suggestions. We appreciate editor and reviewers very much again for your positive and constructive comments and suggestions on our manuscript entitled “Effects of Impact Speeds, Fall Postures and Cortical Thicknesses on Femur Fracture by Incremental Element Deletion Based Finite Element Analysis” (Manuscript ID: materials-1628806). We have studied the comments of both the editor and reviewers carefully and tried our best to revise the paper accordingly. Point-to-point replies are included as below.

Point 2: The title and the majority of the abstract were copied verbatim from a website, which is unethical and unacceptable. To prevent similarities with existing publications, the title and abstract could be altered to the following. Title suggestion: “Incremental Element Deletion Based Finite Element Analysis of the Effects of Impact Speeds, Fall Postures, and Cortical Thicknesses on Femur Fracture”

Response 2: Thank you for your careful review and valuable suggestions. The title was changed to “Incremental Element Deletion Based Finite Element Analysis of the Effects of Impact Speeds, Fall Postures, and Cortical Thicknesses on Femur Fracture” as followed. (Lines 2-4).

Incremental Element Deletion Based Finite Element Analysis of the Effects of Impact Speeds, Fall Postures, and Cortical Thicknesses on Femur Fracture

Point 3: Abstract suggestion: “The proximal femur's numerical simulation could give an effective method for predicting the risk of femoral fracture. However, the majority of existing numerical simulations are static simulations, which do not correctly capture the dynamic properties of bone fractures. On the basis of femoral fracture analysis, a dynamic simulation using incremental element deletion (IED) based finite element analysis (FEA) was developed and compared to XFEM in this study. Mechanical tests were also used to assess it. Different impact speeds, fall postures, and cortical thicknesses were also studied for their implications on fracture types and mechanical responses. The time it took for the crack to shatter was shorter when the speed was higher, and the crack line slid down significantly. The time it took for the crack to shatter was shorter when the speed was higher, and the crack line slid down significantly. The fracture load fell by 27.37 percent when the angle was altered from 15° to 135°, indicating that falling forward was less likely to cause proximal femoral fracture than falling backward. Furthermore, the model with scant cortical bone was susceptible to fracture. This research established a theoretical foundation and mechanism for forecasting the risk of proximal femoral fracture in the elderly.”

Response 3: Thank you for your careful review and valuable suggestions. The title was changed to “The proximal femur's numerical simulation could give an effective method for predicting the risk of femoral fracture. However, the majority of existing numerical simulations are static simulations, which do not correctly capture the dynamic properties of bone fractures. On the basis of femoral fracture analysis, a dynamic simulation using incremental element deletion (IED) based finite element analysis (FEA) was developed and compared to XFEM in this study. Mechanical tests were also used to assess it. Different impact speeds, fall postures, and cortical thicknesses were also studied for their implications on fracture types and mechanical responses. The time it took for the crack to shatter was shorter when the speed was higher, and the crack line slid down significantly. The time it took for the crack to shatter was shorter when the speed was higher, and the crack line slid down significantly. The fracture load fell by 27.37 percent when the angle was altered from 15° to 135°, indicating that falling forward was less likely to cause proximal femoral fracture than falling backward. Furthermore, the model with scant cortical bone was susceptible to fracture. This research established a theoretical foundation and mechanism for forecasting the risk of proximal femoral fracture in the elderly.” as followed. (Lines 20-31).

Abstract: The proximal femur's numerical simulation could give an effective method for pre-dicting the risk of femoral fracture. However, the majority of existing numerical simulations are static, which do not correctly capture the dynamic properties of bone fractures. On the basis of femoral fracture analysis, a dynamic simulation using incremental element deletion (IED) based finite element analysis (FEA) was developed and compared to XFEM in this study. Mechanical tests were also used to assess it. Different impact speeds, fall postures, and cortical thicknesses were also studied for their implications on fracture types and mechanical responses. The time it took for the crack to shatter was shorter when the speed was higher, and the crack line slid down significantly. The fracture load fell by 27.37 percent when the angle was altered from 15° to 135°, indicating that falling forward was less likely to cause proximal femoral fracture than falling backward. Furthermore, the model with scant cortical bone was susceptible to fracture. This study established a theoretical foundation and mechanism for forecasting the risk of proximal femoral fracture in the elderly.

Point 4: I have some suggestions for sentences that were taken directly from websites or documents. It is unethical to use idies from sources without making any alterations, even if references are used for the aforementioned lines.

Response 4: Thank you for your careful review and valuable suggestions. We appreciate editor and reviewers very much again for your positive and constructive comments and suggestions on our manuscript entitled “Effects of Impact Speeds, Fall Postures and Cortical Thicknesses on Femur Fracture by Incremental Element Deletion Based Finite Element Analysis” (Manuscript ID: materials-1628806). We have studied the comments of both the editor and reviewers carefully and tried our best to revise the paper accordingly. Point-to-point replies are included as below.

Point 5: Suggestion for the sentences (page 2, lines 50-55): “Bone fracture may now be studied at both the micro and macro scales because to advances in computer modeling. With a correlation of nearly 90%, linear finite element models have been effectively used to the prediction of the elastic response and fracture load of a human femur [8]. Despite efforts to simulate human femur behavior, fracture patterns have only been modelled using numerical methodologies on a few occasions. Some studies have focused on proximal fracture simulation, with the majority of them yielding short fracture pathways [9-13].”

Response 5: Thank you for your careful review and valuable suggestions. The sentence was changed to“Bone fracture may now be studied at both the micro and macro scales because to advances in computer modeling. With a correlation of nearly 90%, linear finite element models have been effectively used to the prediction of the elastic response and fracture load of a human femur [8]. Despite efforts to simulate human femur behavior, fracture patterns have only been modelled using numerical methodologies on a few occasions. Some studies have focused on proximal fracture simulation, with the majority of them yielding short fracture pathways [9-13].” based on the reviewer's comments as followed. (Lines 52-58).

Bone fracture may now be studied at both the micro and macro scales because to advances in computer modeling. With a correlation of nearly 90%, linear finite element models have been effectively used to the prediction of the elastic response and fracture load of a human femur [8]. Despite efforts to simulate human femur behavior, fracture patterns have only been modelled using numerical methodologies on a few occasions. Some studies have focused on proximal fracture simulation, with the majority of them yielding short fracture pathways [9-13].

Point 6: Suggestion for the sentences (page 2, lines 77-86): Instead of saying “The composite femur has been commonly used in the literature as a simulant of real bone. It is important to emphasise that this kind of specimens is designed to simulate the biomechanical properties of young and healthy femurs [15,16]. These similarities were tested by means of axial compression, bending and torsion tests through the measurement of the corresponding stiffness and ultimate failure strength [15,16]. The use of artificial bone provides advantages for model validation avoiding the variability of properties inherent to biological tissues. Composite bones are useful to develop controlled analysis, due to their homogeneous properties in two distinct zones, smoothed surface and low variability between specimens [15]. The failure modes of these composite models are close to published findings for human bones [16].” I suggest the following wording: “In the literature, the composite femur has been widely employed as a substitute for actual bone. It's vital to note that these specimens are intended to mimic the biomechanical qualities of young, healthy femurs [15,16]. Axial compression, bending, and torsion tests were used to assess these commonalities, with the related stiffness and ultimate failure strength measured [15,16]. Artificial bone has advantages for model validation since it eliminates the variety of properties found in biological tissues. Because of their uniform qualities in two separate zones, smoothed surface, and minimal variability between specimens, composite bones are excellent for developing controlled analysis [15]. These composite models' failure patterns are similar to those reported for human bones [16].”

Response 6: Thank you for your careful review and valuable suggestions. The sentence was changed to “In the literature, the composite femur has been widely employed as a substitute for actual bone. It's vital to note that these specimens are intended to mimic the biomechanical qualities of young, healthy femurs [15,16]. Axial compression, bending, and torsion tests were used to assess these commonalities, with the related stiffness and ultimate failure strength measured [15,16]. Artificial bone has advantages for model validation since it eliminates the variety of properties found in biological tissues. Because of their uniform qualities in two separate zones, smoothed surface, and minimal variability between specimens, composite bones are excellent for developing controlled analysis [15]. These composite models' failure patterns are similar to those reported for human bones [16].” based on the reviewer's comments as followed. (Lines 79-87).

Five newly designed fourth-generation composite femurs provided by Sawbones (Sawbones, Pacific Research Laboratories, Vashon, USA) were used to investigate fracture behaviors. In the literature [15,16], the composite femur has been widely employed as a substitute for actual bone. It's vital to note that these specimens are intended to mimic the biomechanical qualities of young, healthy femurs. Axial compression, bending, and torsion tests were used to assess these commonalities, with the related stiffness and ultimate failure strength measured [15,16]. Artificial bone has advantages for model validation since it eliminates the variety of properties found in biological tissues. Because of their uniform qualities in two separate zones, smoothed surface, and minimal variability between specimens, composite bones are excellent for developing controlled analysis [15]. These composite models' failure patterns are similar to those reported for human bones [16]. The composite femurs were scanned by computerized tomography (CT) at first. The entire composite femur was scanned by CT with a slice thickness of 2.5 mm and a pixel width of 0.938 mm (GE MEDICAL SYSTEMS/LightSpeed 16 Computed Tomography Scanner System, 80 kV, 443.52 mAs, 512 × 512 matrix, 52 images). IED based FEA on femoral fracture analysis was developed and compared with XFEM, which was also evaluated by mechanical tests. The strains of 8 positions on the surfaces of composite femurs were collected in elastic tests (as shown in Fig. 1a), Fig. 1b), Fig. 1c), Fig. 1d)). The fracture loads and crack propagation paths of composite femurs were collected in fracture tests.

Point 7: Suggestion for the sentences (page 4, lines 120-123): Instead of saying “The loads were applied to the femur in the stance loading direction (Fig. 2a). This position is the common configuration analysed in the literature because it corresponds to the physiological scheme of a standingup human position[19].” I suggest the following wording: “The loads were applied to the femur in the direction of stance loading (Fig. 2a). Because it matches to the physiological plan of a standing-up human position, this is the most commonly studied configuration in the literature.”

Response 7: Thank you for your careful review and valuable suggestions. The sentence was changed to“The loads were applied to the femur in the direction of stance loading (Fig. 2a). Because it matches to the physiological plan of a standing-up human position, this is the most commonly studied configuration in the literature.” based on the reviewer's comments as followed. (Lines 120-123).

The composite femur was tested on a 10 kN universal hydraulic testing machine (Instron 8801, load cell 10 kN). As shown in Fig. 1c, the distal end of the composite femur was inserted into the dental powder mixture to maintain the composite femur with α angle of 10° and β angle of 15°, and it was fixed for about 30 min - 60 min. The loads were applied to the femur in the direction of stance loading (Fig. 2). Because it matches to the physiological plan of a standing-up human position, this is the most commonly studied configuration in the literature [19]. During the test, 8 strain gauges (Qinhuangdao Aifu Te Electronic Technology Co., Ltd., BX-120) were used to measure the strains at 8 different positions on the surfaces of composite femurs. The specific locations where the strain gauges were pasted, as shown in Fig.1e. To maintain quasi-static conditions, the load was increased up to the different values (250 N, 500 N, and 750 N with an actuator speed of 0.3 mm/s). This test confirmed the linear elastic behavior of the femur, as described by another author [20].

Point 8: Suggestion for the sentences (page 4, lines 126-129): Instead of saying “Load was increased up to the different values considered (250 N, 500 N and 750 N with an actuator speed of 0.3 mm/s in order to keep quasi-static conditions). The femur linear elastic behaviour was verified through this test, as reported by other author [20].” I suggest the following wording: “To maintain quasi-static conditions, the load was increased up to the different values evaluated (250 N, 500 N, and 750 N with an actuator speed of 0.3 mm/s). This test confirmed the linear elastic behavior of the femur, as described by another author [20].”

Response 8: Thank you for your careful review and valuable suggestions. The sentence was changed to“To maintain quasi-static conditions, the load was increased up to the different values evaluated (250 N, 500 N, and 750 N with an actuator speed of 0.3 mm/s). This test confirmed the linear elastic behavior of the femur, as described by another author [20].” based on the reviewer's comments as followed. (Lines 126-129).

The composite femur was tested on a 10 kN universal hydraulic testing machine (Instron 8801, load cell 10 kN). As shown in Fig. 1c, the distal end of the composite femur was inserted into the dental powder mixture to maintain the composite femur with α angle of 10° and β angle of 15°, and it was fixed for about 30 min - 60 min. The loads were applied to the femur in the direction of stance loading (Fig. 2). Because it matches to the physiological plan of a standing-up human position, this is the most commonly studied configuration in the literature [19]. During the test, 8 strain gauges (Qinhuangdao Aifu Te Electronic Technology Co., Ltd., BX-120) were used to measure the strains at 8 different positions on the surfaces of composite femurs. The specific locations where the strain gauges were pasted, as shown in Fig.1e. To maintain quasi-static conditions, the load was increased up to the different values (250 N, 500 N, and 750 N with an actuator speed of 0.3 mm/s). This test confirmed the linear elastic behavior of the femur, as described by another author [20].

Point 9: Suggestion for the sentences (pages 4-5, lines 155-158): Instead of saying “The boundary conditions and the loading state are set as close as possible to the conditions imposed during the experiments. The distal part of the diaphysis is constrained to the testing rig simulating the embedment in surgical cement and the load was applied in the proper direction similar to the configuration used in the experiments.” I suggest the following wording: “The boundary conditions and loading state are set to be as close to the experimental conditions as possible. The distal part of the diaphysis is restrained to the testing rig, which simulates surgical cement embedment, and the force is imparted in the right direction, as in the tests.”

Response 9: Thank you for your careful review and valuable suggestions. The sentence was changed to“The boundary conditions and loading state are set to be as close to the experimental conditions as possible. The distal part of the diaphysis is restrained to the testing rig, which simulates surgical cement embedment, and the force is imparted in the right direction, as in the tests.” based on the reviewer's comments as followed. (Lines 155-158).

The boundary conditions and loading state are set to be as close to the experimental conditions as possible. The distal part of the diaphysis is restrained to the testing rig, which simulates the surgical cement embedment, and the force is imparted in the right direction, as in the tests. The distal end of the composite femur was fixed. α angle of 10° and β angle of 15° were used to simulate the fall posture (as shown in Fig. 2c). The loading was exerted on a spherical region with a 35 mm diameter on the femoral head (this is in agreement with the numerical models developed by another author [20]), and the fracture process was simulated. The maximum principal stress, displacement at point A in Fig. 1a, and the running time of the program were recorded in the models with 7 different element sizes mentioned above. Finally, the results of FEA were com-pared with the results of mechanical tests.

Point 10: Suggestion for the sentences (page 7, lines 240-244): Instead of saying “The developed models reproduce the mechanical behaviour of the loading of composite femurs defined in the experiments. In the elastic regime, femurs were instrumented with 8 strain gages. Since the three measurements were recorded at 3 different loading steps (250 N, 500 N, 750 N), a total of 24 strain datas were registered for each femur to validate the corresponding numerical model.” I suggest the following wording: “The generated models accurately represent the mechanical behavior of composite femur loading as seen in the experiments. Femurs were fitted with eight strain gages in the elastic regime. Because the three measurements were taken at three different loading levels (250 N, 500 N, and 750 N), each femur had a total of 24 strain data points to validate the numerical model.”

Response 10: Thank you for your careful review and valuable suggestions. The sentence was changed to “The generated models accurately represent the mechanical behavior of composite femur loading as seen in the experiments. Femurs were fitted with eight strain gages in the elastic regime. Because the three measurements were taken at three different loading levels (250 N, 500 N, and 750 N), each femur had a total of 24 strain data points to validate the numerical model.” based on the reviewer's comments as followed. (Lines 261-266).

From the above results in section 3.2,1, we calculated the fit curve of the IED based FEA. The generated models accurately represent the mechanical behavior of composite femur loading as seen in the experiments. Femurs were fitted with eight strain gages in the elastic regime. Because the three measurements were taken at three different loading levels (250 N, 500 N, and 750 N), each femur had a total of 24 strain data points to validate the numerical model. It can be seen that numerical model predictions were correlated (R2 = 0.9173) to the experimental values for 24 strains (8 strain gages for each load case) at different loads (250 N, 500 N, 750 N) (as shown in Fig. 6). The values for the mean, standard deviation in the experimental and numerical results were respectively, 52.85, 246.27, and 64.37, 237.52. There was a significant correlation between the experi-mental and numerical strain (COV=58196>0, 95%CI, 0.79-1.05, p<0.0001).

Point 11: Suggestion for the sentences (page 8, lines 254-260): Instead of saying “Regarding the predicted fracture location and early stages of crack growth shown in Fig. 6a-b, it can be seen that it is similar to the failure region observed in Fig. 6c. Moreover, the numerical results showed the same trends obtained by other authors also with composite femurs [27]. Concerning the fracture loading, the experimental test yielded a maximum load of 6573±10 N. The fracture test occurred at the neck, as it is also found in the works by other author [28]. Our numerical simulations also predict the failure at this zone; see the fracture path calculated for the stance loading case in Fig. 6d..” I suggest the following wording: “It can be seen in Fig. 6a-b that the projected fracture location and early phases of crack propagation are identical to the failure zone seen in Fig. 6c. Furthermore, the numerical results confirmed the same tendencies found by other researchers using composite femurs [27]. The experimental test revealed a maximum load of 6573±10 N in terms of fracture loading. The fracture test was performed on the neck, as described by other authors [28]. Our numerical simulations similarly anticipate failure in this zone; see Fig. 6d for the projected fracture route for the stance loading case.”

Response 11: Thank you for your careful review and valuable suggestions. The sentence was changed to “It can be seen in Fig. 6a-b that the projected fracture location and early phases of crack propagation are identical to the failure zone seen in Fig. 6c. Furthermore, the numerical results confirmed the same tendencies found by other researchers using composite femurs [27]. The experimental test revealed a maximum load of 6573±10 N in terms of fracture loading. The fracture test was performed on the neck, as described by other authors [28]. Our numerical simulations similarly anticipate failure in this zone; see Fig. 6d for the projected fracture route for the stance loading case.” based on the reviewer's comments as followed. (Lines 242-248).

The relative error between the simulated and experimental fracture load calculated by Eq.3 were listed in Table 1. The crack propagation path of the fracture test was shown in Fig. 5c, while the corresponding numerical results were shown Fig. 5a-b for each method evaluated. It can be seen in Fig. 5a-b that the projected fracture location and early phases of crack propagation are identical to the failure zone seen in Fig. 5c. Fur-thermore, the numerical results also confirmed by other researchers using composite femurs [27]. The experimental test revealed a maximum load of 6573±10 N in terms of fracture loading. The fracture test was performed on the neck, as described by other authors [28]. Our numerical simulations similarly anticipate failure in this zone; see Fig. 5d for the projected fracture route for the stance loading case. It was found that the results include relative error and crack propagation path of IED based FEA were closest to the results of mechanical tests. Therefore, in the following study, IED based FEA was selected for simulation of different impact speeds, fall postures, and cortical thicknesses on fracture types and mechanical responses.

Point 12: The first paragraph (page 12, lines 371-379) in the discussion section was copied verbatim from a website and needs to be changed with the authors' own words. The following is an example of a suggestion: “IED-based FEA results in longer fracture trajectories than XFEM. Due to convergence issues, XFEM produced unsatisfactory results for long crack trajectories. As a result, pathways generated using an IED-based FEA showed good convergence behavior, resulting in extended trajectories. Because each increment of the fracture growth represents a fresh simulation, this technique avoids convergence issues. When comparing the two procedures, element elimination has more issues due to the presence of distorted elements, which might cause the numerical process to slow down. As a result, the IED-based FEA technique produces the best results in terms of convergence and fracture path length, and it can be applied to a variety of loading scenarios.”

Response 12: Thank you for your careful review and valuable suggestions. The sentence was changed to “IED-based FEA results in longer fracture trajectories than XFEM. Due to convergence issues, XFEM produced unsatisfactory results for long crack trajectories. As a result, pathways generated using an IED-based FEA showed good convergence behavior, resulting in extended trajectories. Because each increment of the fracture growth represents a fresh simulation, this technique avoids convergence issues. When comparing the two procedures, element elimination has more issues due to the presence of distorted elements, which might cause the numerical process to slow down. As a result, the IED-based FEA technique produces the best results in terms of convergence and fracture path length, and it can be applied to a variety of loading scenarios.” based on the reviewer's comments as followed. (Lines 374-382).

IED-based FEA results in longer fracture trajectories than XFEM. Due to conver-gence issues, XFEM produced unsatisfactory results for long crack trajectories. As a result, pathways generated using an IED-based FEA showed good convergence be-havior, resulting in extended trajectories. Because each increment of the fracture growth represents a fresh simulation, this technique avoids convergence issues. When com-paring the two procedures, element elimination has more issues due to the presence of distorted elements, which might cause the numerical process to slow down. As a result, the IED-based FEA technique produces the best results in terms of convergence and fracture path length, and it can be applied to a variety of loading scenarios.

Point 13: The usage of a 3D tetrahedral mesh in FE analysis is discussed on page 4, lines 146-154, but no element specifics are stated. As a result, please include any element details used in the FE analysis.

Response 13: Thank you for your careful review and valuable suggestions. The element details used in the FE analysis were added in section 3.1, such as the element numbers in Figure 4a). (Lines 222-232).

IED-based FEA results in longer fracture trajectories than XFEM. Due to conver-gence issues, XFEM produced unsatisfactory results for long crack trajectories. As a result, pathways generated using an IED-based FEA showed good convergence be-havior, resulting in extended trajectories. Because each increment of the fracture growth represents a fresh simulation, this technique avoids convergence issues. When com-paring the two procedures, element elimination has more issues due to the presence of distorted elements, which might cause the numerical process to slow down. As a result, the IED-based FEA technique produces the best results in terms of convergence and fracture path length, and it can be applied to a variety of loading scenarios.

Point 14: The regression analysis findings are a little dubious. As a result, I recommend including values for the mean, standard deviation, and COV in the experimental results.

Response 14: Thank you for your careful review and valuable suggestions. The values for the mean, standard deviation, and COV in the experimental results were added as followed. (Lines 268-271).

From the above results in section 3.2,1, we calculated the fit curve of the IED based FEA. The generated models accurately represent the mechanical behavior of composite femur loading as seen in the experiments. Femurs were fitted with eight strain gages in the elastic regime. Because the three measurements were taken at three different loading levels (250 N, 500 N, and 750 N), each femur had a total of 24 strain data points to validate the numerical model. It can be seen that numerical model predictions were correlated (R2 = 0.9173) to the experimental values for 24 strains (8 strain gages for each load case) at different loads (250 N, 500 N, 750 N) (as shown in Fig. 6). The values for the mean, standard deviation in the experimental and numerical results were respectively, 52.85, 246.27, and 64.37, 237.52. There was a significant correlation between the experimental and numerical strain (COV=58196>0, 95%CI, 0.79-1.05, p<0.0001).

Point 15: Furthermore, the confidence interval (e.g., for 95% and 99%) values of the experiments will improve the results' reliability as well as the paper's quality.

Response 15: Thank you for your careful review and valuable suggestions. In order to improve the results' reliability as well as the paper's quality, the confidence interval (e.g., for 95%) values of the experiments were added as followed. (Lines 268-271).

From the above results in section 3.2,1, we calculated the fit curve of the IED based FEA. The generated models accurately represent the mechanical behavior of composite femur loading as seen in the experiments. Femurs were fitted with eight strain gages in the elastic regime. Because the three measurements were taken at three different loading levels (250 N, 500 N, and 750 N), each femur had a total of 24 strain data points to validate the numerical model. It can be seen that numerical model predictions were correlated (R2 = 0.9173) to the experimental values for 24 strains (8 strain gages for each load case) at different loads (250 N, 500 N, 750 N) (as shown in Fig. 6). The values for the mean, standard deviation in the experimental and numerical results were respectively, 52.85, 246.27, and 64.37, 237.52. There was a significant correlation between the experimental and numerical strain (COV=58196>0, 95%CI, 0.79-1.05, p<0.0001).

Some minor typos should be corrected.

For instance:

Point 16: Page 1, line 43, replace “on” by “in”

Response 16: Thank you for your careful review and valuable suggestions. The “on” was changed to “in” as followed. (Line 45).

Compared with the elderly, falls, which occur on young and middle-aged people in the same way and with the same external factors, rarely cause similar fractures.

Point 17: Page 2, line 50, replace “modelling” by “modeling” because the spelling of modelling (with double l) is a non-American variant. For consistency, consider replacing it with the American English spelling.

Response 17: Thank you for your careful review and valuable suggestions. The “modelling” was changed to “modeling” as followed. (Line53).

Bone fracture may now be studied at both the micro and macro scales because to advances in computer modeling.

Point 18: Page 2, line 64, replace the second “simulations” by “ones” or delete that word.

Response 18: Thank you for your careful review and valuable suggestions. The “simulations” was changed to “ones” as followed. (Line 66).

The extant numerical simulations are mostly static ones, which are difficult to represent the dynamic crack expansion process [8,14].

Point 19: Page 2, line 78, replace “specimens” by “specimen” (without “s”)

Response 19: Thank you for your careful review and valuable suggestions. The sentence was changed to “t's vital to note that these specimens are intended to mimic the bio-mechanical qualities of young, healthy femurs.”. (Lines 80-81).

In the literature [15,16], the composite femur has been widely employed as a substitute for actual bone. It's vital to note that these specimens are intended to mimic the bio-mechanical qualities of young, healthy femurs.

Point 20: Page 4, line 126, Before “load” please use “The”.

Response 20: Thank you for your careful review and valuable suggestions. The “The” was added before “load” as followed. (Line 126).

To maintain quasi-static conditions, the load was increased up to the different values (250 N, 500 N, and 750 N with an actuator speed of 0.3 mm/s). This test confirmed the linear elastic behavior of the femur, as described by another author [20].

Point 21: Page 4, line 129, replace “author” by “authors”

Response 21: Thank you for your careful review and valuable suggestions. The “author” was changed to “another author” as followed. (Line129).

To maintain quasi-static conditions, the load was increased up to the different values (250 N, 500 N, and 750 N with an actuator speed of 0.3 mm/s). This test confirmed the linear elastic behavior of the femur, as described by another author [20].

Point 22: Page 4, line 137, add “the” before “simulation”

Response 22: Thank you for your careful review and valuable suggestions. The “the” was added before “simulation” as followed. (Line 137).

The loading conditions of the simulation were exactly the same as the mechanical tests.

Point 23: Page 4, line 146, add “the” before “element”

Response 23: Thank you for your careful review and valuable suggestions. The “the” was added before “element” as followed. (Line146).

The element size of the FE model is very important for the accuracy of the result [21].

Point 24: Page 4, line 146, the “the” in front of the “both” should be deleted.

Response 24: Thank you for your careful review and valuable suggestions. The “the” was deleted in front of the “both” as followed. (Line148).

In this study, both 3D solid parts were divided into 3D tetrahedral mesh, and the element sizes were set to 0.5 mm, 0.75 mm, 1 mm, 1.5 mm, 2 mm, 2.5 mm and 3 mm

Point 25: Page 5, line 160, Before “loading” please use “The”.

Response 25: Thank you for your careful review and valuable suggestions. The “the” was added before “loading” as followed. (Line159).

The loading was exerted on a spherical region with a 35 mm diameter on the femoral head (this is in agreement with the numerical models developed by another author [20])

Point 26: Page 5, line 162, replace “author” by “authors”

Response 26: Thank you for your careful review and valuable suggestions. The “author” was changed to “authors” as followed. (Line 161).

The loading was exerted on a spherical region with a 35 mm diameter on the femoral head (this is in agreement with the numerical models developed by another author [20])

Point 27: Page 5, line 171, use comma (,) before “and”

Response 27: Thank you for your careful review and valuable suggestions. The comma (,) was added before “and”. (Line 170).

In this study, Young’s modulus, bone density, and Poisson’s ratio were assigned to the model according to the study by Marco et al [22]. An initial increment "n" of the load was set, and the principal strain of the composite femur was calculated.

Point 28: Page 5, lines 171 and 172, replace “An initial increment n of load was set….” by “An initial increment "n" of the load was set…”

Response 28: Thank you for your careful review and valuable suggestions. The sentence “An initial increment n of load was set….” was changed to “An initial increment "n" of the load was set…” as followed. (Lines 172-173).

In this study, Young’s modulus, bone density, and Poisson’s ratio were assigned to the model according to the study by Marco et al [22]. An initial increment "n" of the load was set, and the principal strain of the composite femur was calculated.

Point 29: Page 5, line 177, Before “principal” please use “the”.

Response 29: Thank you for your careful review and valuable suggestions. The “the” was added before “principal” as followed. (Line 177).

During the loading process, the principal strain of each element in the composite femur model was compared, when it exceeded the failure strain, Young’s modulus was reduced to the minimum value (E = 1 MPa) in order to reduce the element stiffness to a negligible value.

Point 30: Page 5, line 178, the “the” in front of the “Young’s modulus” should be deleted.

Response 30: Thank you for your careful review and valuable suggestions. The “the” was deleted in front of the “Young’s modulus” as followed. (Line 179).

During the loading process, the principal strain of each element in the composite femur model was compared, when it exceeded the failure strain, Young’s modulus was reduced to the minimum value (E = 1 MPa) in order to reduce the element stiffness to a negligible value.

Point 31: Page 6, line 187, Before “XFEM” please use “the”.

Response 31: Thank you for your careful review and valuable suggestions. The “the” was added before “XFEM” as followed. (Line 186).

By using the XFEM module in Abaqus/Standard, the virtual crack closure technology (VCCT) was used to simulate the crack propagation process.

Point 32: Page 6, line 191, use comma (,) before “and”

Response 32: Thank you for your careful review and valuable suggestions. The comma (,) was added before “and”. (Line 190).

The Young’s modulus, bone density, Poisson’s ratio, and compressive failure strain required in this method according to the study by Marco et al [22].

Point 33: Page 6, line 194, the comma (,) in front of the “and” (second one) should be deleted.

Response 33: Thank you for your careful review and valuable suggestions. The comma (,) in front of the “and” (second one) was deleted. (Line 193).

The critical energy value (GC), necessary for XFEM to predict the start of crack propa-gation, which was estimated from the fracture toughness KC, as shown in Eq. (1), and the fracture toughness KC was related to bone density and could be obtained by Eq. (2). The following expressions determine these relationships [24].

Point 34: Page 6, line 220, add “was” before “fractured”

Response 34: Thank you for your careful review and valuable suggestions. The “was” was added before “fractured” as followed. (Line 219).

The dynamic pressure loading was applied at a speed of 0.3 mm/s on a 35 mm diameter spherical region above the femoral head. It was performed until the composite femur was fractured.

Point 35: Page 7, line 243, replace “datas” by “data”.

Response 35: Thank you for your careful review and valuable suggestions. The “datas” was changed to “data” as followed. (Line 265).

From the above results in section 3.2,1, we calculated the fit curve of the IED based FEA. The generated models accurately represent the mechanical behavior of composite femur loading as seen in the experiments. Femurs were fitted with eight strain gages in the elastic regime. Because the three measurements were taken at three different loading levels (250 N, 500 N, and 750 N), each femur had a total of 24 strain data points to validate the numerical model.

Point 36: Page 8, line 251, Before “relative” please use “The”.

Response 36: Thank you for your careful review and valuable suggestions. The “The” was added before “relative” as followed. (Line 239).

The relative error between the simulated and experimental fracture load calculated by Eq.3 were listed in Table 1.

Point 37: Page 8, line 252, add “the” before “fracture”

Response 37: Thank you for your careful review and valuable suggestions. The “the” was added before “fracture” as followed. (Line 240).

The crack propagation path of the fracture test was shown in Fig. 5c

Point 38: Page 8, line 259, replace “author” by “authors”

Response 38: Thank you for your careful review and valuable suggestions. The “author” was changed to “authors” as followed. (Line246).

The experimental test revealed a maximum load of 6573±10 N in terms of fracture loading. The fracture test was performed on the neck, as described by other authors [28].

Point 39: Page 9, line 279, replace “of” by “in”

Response 39: Thank you for your careful review and valuable suggestions. The “of” was changed to “in” as followed. (Line285).

It could be seen that when the impact speed exceeded a certain range, the greater of impact speed, the longer of crack length in the composite femur.

Point 40: Page 9, line 283, replace “thes” by “the”

Response 40: Thank you for your careful review and valuable suggestions. The “thes” was changed to “the” as followed. (Line 286).

Fig. 7d showed the effects of impact speeds on the instantaneous speeds which meant the speed of crack propagation of composite proximal femoral fractures.

Point 41: Page 9, line 284, please add “that” before “with”

Response 41: Thank you for your careful review and valuable suggestions. The “that” was added before “with” as followed. (Line 287).

It could be seen that with the increase of impact speed, the instantaneous speed at the time of fracture became greater, and the greater harm to the human body.

Point 42: Page 9, line 285, please add “the” before “human”

Response 42: Thank you for your careful review and valuable suggestions. The “the” was added before “human” as followed. (Line 288).

It could be seen that with the increase of impact speed, the instantaneous speed at the time of fracture became greater, and the greater harm to the human body.

Point 43: Page 9, line 290, the “the” in front of the “four” should be deleted.

Response 43: Thank you for your careful review and valuable suggestions. The “the” in front of the “four” was deleted as followed. (Line 293).

It could be seen that all four impact speeds caused femoral neck fractures, and all cracks started from the femoral neck.

Point 44: Page 10, line 301, use comma (,) before first “and”

Response 44: Thank you for your careful review and valuable suggestions. The comma (,) was added before “and”. (Line 304).

When β = 0°, 15°, 30°, and 45°, the relationships between the α angle and the maximum principal stress, maximum principal strain, and fracture time were analyzed.

Point 45: Page 10, line 302, use comma (,) before “and”

Response 45: Thank you for your careful review and valuable suggestions. The comma (,) was added before “and”. (Line 305).

When β = 0°, 15°, 30°, and 45°, the relationships between the α angle and the maximum principal stress, maximum principal strain, and fracture time were analyzed.

Point 46: Page 10, line 340, delete one of the “the”

Response 46: Thank you for your careful review and valuable suggestions. One of the “the” was deleted as followed. (Line 343).

When α angle did not change, as β increased, the fracture axis moved downward (Fig. 9d-f) from the top of the composite femur neck to the bottom of the composite femur neck.

Point 47: Page 10, line 360, replace “resulted” by “resulting”

Response 47: Thank you for your careful review and valuable suggestions. The “resulted” was changed to “resulting” as followed. (Line 363).

As the cortical bone became thinner, the fracture line gradually moved from the composite femoral neck to the intertrochanter, resulting in different types of composite proximal femoral fractures, which indicated that the thickness of cortical bone was an important factor affecting the types of composite proximal femoral fractures.

Point 48: Page 12, line 374, the comma (,) in front of the “because” should be deleted.

Response 48: Thank you for your careful review and valuable suggestions. The comma (,) in front of the “because” should be deleted as followed. (Line 377).

IED-based FEA results in longer fracture trajectories than XFEM. Due to conver-gence issues, XFEM produced unsatisfactory results for long crack trajectories. As a result, pathways generated using an IED-based FEA showed good convergence be-havior, resulting in extended trajectories. Because each increment of the fracture growth represents a fresh simulation, this technique avoids convergence issues. When com-paring the two procedures, element elimination has more issues due to the presence of distorted elements, which might cause the numerical process to slow down. As a result, the IED-based FEA technique produces the best results in terms of convergence and fracture path length, and it can be applied to a variety of loading scenarios.

Point 49: Page 12, line 384, replace “literatures” by “literature”

Response 49: Thank you for your careful review and valuable suggestions. The “literatures” was changed to “literature” as followed. (Line 387).

As the impact speed increased, the complexity and roughness of fracture cracks also increased, which were consistent with the results in the literature [26, 31].

Point 50: Page 12, line 384, please add “that” before “when”

Response 50: Thank you for your careful review and valuable suggestions. The “on” was changed to “in” as followed. (Line 387).

Ren et al. [32] pointed out that when the load continued to increase beyond the strain range, a large area of bone would be fractured.

Point 51: Page 12, line 387, please add “s” after “side”

Response 51: Thank you for your careful review and valuable suggestions. The “s” was added after “side” as followed. (Line390).

In this study, cracks first appeared on the outer and upper sides.

Point 52: By the way, the reference numbers are written next to the final words of each phrase. Before the reference numbers, a gap should be left.

Response 1: Thank you for your careful review and valuable suggestions. The reference in this paper was corrected. The reference numbers were written next to the final words of each phrase. And before the reference numbers, a gap was left.

Point 53: The conclusion should be written in bullet points with only the most important findings.

Response 53: Thank you for your careful review and valuable suggestions. The conclusion has been rewritten with only the most important findings as followed. (Lines 475-490).

In this study, IED based FEA was developed and compared with XFEM. Moreover, the effects of different impact speeds, fall postures, and cortical thicknesses on fracture types and mechanical responses were investigated by IED based FEA. IED based FEA was found to better simulate the occurrence and development of composite proximal femoral fracture than XFEM with comparisons of mechanical tests. It could well predict the effects of different fall conditions on fracture types and mechanical responses. When the speed was faster, the time of fracture was shorter, and the crack line moved down significantly; When α angle changed from 15° to 135°, the fracture load decreased by 27.37% indicated that falling forward was less likely to cause proximal femoral fracture compared with falling backward; Besides, the model with thin cortical bone was prone to fracture, and when the entire model was cancellous bone, the femoral head was fractured. The study conducted a comprehensive theoretical analysis of proximal fem-oral fractures, which may provide sufficient theoretical support in the development of prevention methodology of femoral fractures. Through this technique, it is possible to simulate long fracture paths, which is important when fracture morphology is studied, since different fracture morphologies must be treated with distinct surgical treatments.

Point 54: The following studies must be evaluated and cited because the authors have used some information from them directly or indirectly, or because there are relevant research that the authors have not discussed or cited. The following are the most important studies:

“The effect of load direction on the structure capacity of human proximal femur during falling” in Applied Mechanics and Materials, vol. 137, pp. 7-11, 2011

Response 54-1: Thank you for your careful review and valuable suggestions. The study mentioned above was evaluated and cited as followed. (Line 410-417).

Ford and Kryak et al. [37, 38] also reached a conclusion that was basically consistent with this study, which pointed out that 26% reduction in load capacity was equivalent to the result of bone mineral density loss as 25 years after the age of 65. The structural capacity of the proximal femur, like any other structure, is determined by the applied loads, which can vary depending on the direction of impact during a fall. Our findings demonstrate the independent contribution of fall mechanics to hip fracture risk by identifying a fall aspect (the direction of impact) that is an important determinant of fall severity. This is also consistent with the results of Fu et al [39].

“Simulation of creep in non-homogenous samples of human cortical bone” in Computer Methods in Biomechanics and Biomedical Engineering, vol. 15, pp. 1121-1128, 2012

Response 54-2: Thank you for your careful review and valuable suggestions. The study mentioned above was evaluated and cited as followed. (Lines 435-439).

Ertas et al. [46] simulated 10 heterogeneous samples using an empirically validated creep strain accumulation model to determine the relationship between steady-state creep rate, applied load, and microstructure. They state that impact velocity is one of the factors that contribute to the mechanism of viscoelastic mechanical properties of human cortical bone, and that it is important for predicting bone response to creep and fatigue loading. They also pointed out that creep and fatigue behavior of cancellous bone are important in senile fractures, bone remodeling, and implant subsidence. Mechanical tests on cancellous bone samples have been used to investigate permanent deformation over time under creep and fatigue loading. They also investigated the creep behavior of porcine cancellous bone, finding strong relationships between applied stress and both time-to-failure and steady-state creep rate, which is consistent with our findings [47].

“Creep simulation of a micro-CT based finite element model of porcine cancellous bone” in Summer Engineering Conference, 54587, pp. 285-286, 2011

Response 54-3: Thank you for your careful review and valuable suggestions. The study mentioned above was evaluated and cited as followed. (Lines 440-445).

Ertas et al. [46] simulated 10 heterogeneous samples using an empirically validated creep strain accumulation model to determine the relationship between steady-state creep rate, applied load, and microstructure. They state that impact velocity is one of the factors that contribute to the mechanism of viscoelastic mechanical properties of human cortical bone, and that it is important for predicting bone response to creep and fatigue loading. They also pointed out that creep and fatigue behavior of cancellous bone are important in senile fractures, bone remodeling, and implant subsidence. Mechanical tests on cancellous bone samples have been used to investigate permanent deformation over time under creep and fatigue loading. They also investigated the creep behavior of porcine cancellous bone, finding strong relationships between applied stress and both time-to-failure and steady-state creep rate, which is consistent with our findings [47].

Point 55: Finally, while the work is well-written in general, it does contain some grammatical and typographical problems. Before resubmitting the manuscript, it is suggested that the authors reread it again.

Response 55: Thank you for your careful review and valuable suggestions. We would like to express our sincere thanks again to you for the constructive and positive comments on our manuscript entitled “Effects of Impact Speeds, Fall Postures and Cortical Thicknesses on Femur Fracture by Incremental Element Deletion Based Finite Element Analysis” (Manuscript ID: materials-1628806). We have optimized the article again this time, the grammar errors, sentence structure and sentence fragments, etc. had been revised in revised manuscript.

Round 2

Reviewer 2 Report

The authors did their best to answer all questions. The paper can then be accepted for publication. 

Reviewer 3 Report

The paper is now complete and ready to be published.